# A Closer Look at Multimodal Representation Collapse

**Abhra Chaudhuri** [1]  **Anjan Dutta** [2]  **Tu Bui** [1]  **Serban Georgescu** [1]

## Abstract

We aim to develop a fundamental understanding of modality collapse, a recently observed empirical phenomenon wherein models trained for multimodal fusion tend to rely only on a subset of the modalities, ignoring the rest. We show that modality collapse happens when noisy features from one modality are entangled, via a shared set of neurons in the fusion head, with predictive features from another, effectively masking out positive contributions from the predictive features of the former modality and leading to its collapse. We further prove that cross-modal knowledge distillation implicitly disentangles such representations by freeing up rank bottlenecks in the student encoder, denoising the fusion-head outputs without negatively impacting the predictive features from either modality. Based on the above findings, we propose an algorithm that prevents modality collapse through explicit basis reallocation, with applications in dealing with missing modalities. Extensive experiments on multiple multimodal benchmarks validate our theoretical claims. Project page: https://abhrac.github.io/mmcollapse/.

## 1. Introduction

A number of recent works in the multimodal learning literature have observed that models that aim to learn a fusion of several modalities often end up relying only on a subset of them (Javaloy et al., 2022; Wu et al., 2024). This phenomenon, termed as *modality collapse*, has been empirically observed across a diverse range of fusion strategies (Javaloy et al., 2022; Ma et al., 2022; Zhang et al., 2022; Zhou et al., 2023; Wu et al., 2024), and has serious implications for the scenario when certain modalities can go missing at test time (You et al., 2020; Ma et al., 2022; Wu et al., 2024). If a

---
[1]Fujitsu Research of Europe [2]University of Surrey. Correspondence to: Abhra Chaudhuri <abhra.chaudhuri@fujitsu.com>.

*Proceedings of the 42nd International Conference on Machine Learning*, Vancouver, Canada. PMLR 267, 2025. Copyright 2025 by the author(s).

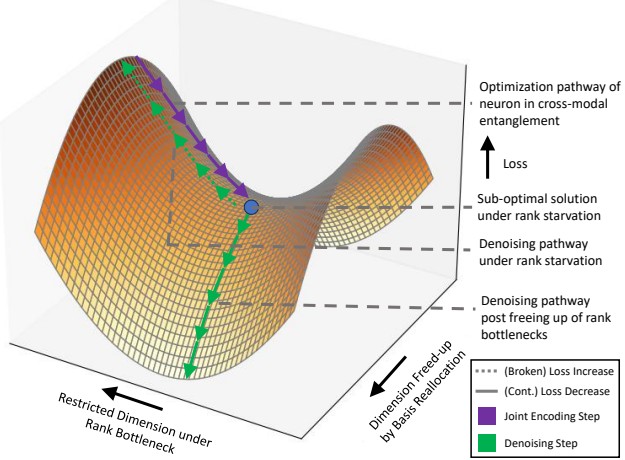

*Figure 1.* When noisy features of one modality exist in entanglement with predictive features of another in the fusion head (the probability of which increases with the number of modalities), it results in a sub-optimal solution wherein the predictive value of the former modality is diminished by the inevitable existence of noisy features. Freeing up rank bottlenecks allows for the denoising of such features along independent dimensions without affecting the latter modality, while simultaneously allowing the predictive features of the former modality to contribute to loss reduction.

model is reliant only on a subset of modalities and if it is those that specifically go missing at test time, the model could end up completely non-functional. Although there have been several attempts at mitigating modality collapse based on a priori conjectures about what might be causing them, such as conflicting gradients (Javaloy et al., 2022) or interactions between data distributions and the fusion strategy (Ma et al., 2022), to the best of our knowledge, there have been no prior efforts towards developing a bottom-up understanding of the underlying learning-theoretic phenomena at play.

We aim to bridge this gap by developing a mechanistic theory of multimodal feature encoding that is agnostic of the specific choice of fusion strategy (see Task Setup in Section 3). We start by showing that modality collapse arises as a result of an unintended entanglement among the noisy and the predictive features of different modalities through a shared set of polysemantic neurons (Elhage et al., 2022), which we observe, via Lemma 1, to increase quadratically

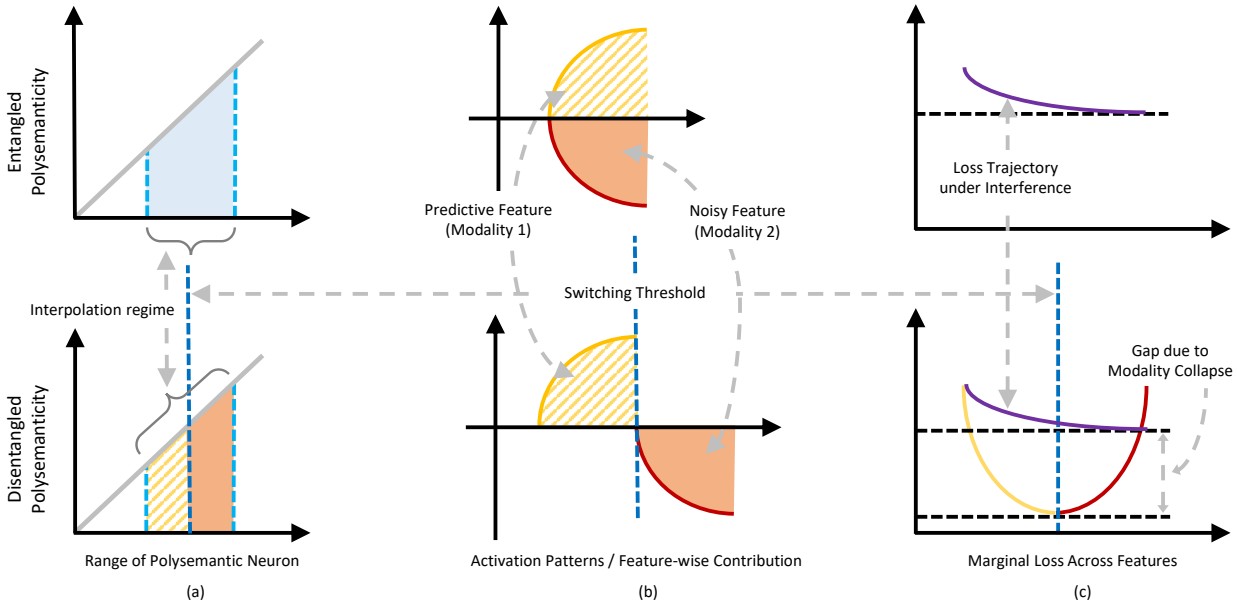

*Figure 2.* Polysemanticity with and without feature interference between noisy and predictive features. All horizontal axes correspond to the value of the weight of the polysemantic neuron. For (b) and (c), the vertical axes correspond to the title of the columns. When the predictive features of modality 1 (M1) and the noisy features of modality 2 (M2) activate along the same region in the interpolation regime of the same neuron (a - top), it prevents the predictive features of M2 from contributing to loss minimization (c - top) because of the unintended inclusion of and the unwanted interference from the noisy features of M2 (b - top), leading to its collapse. However, when they are disentangled (here, by being mapped to disjoint sub-regimes in the weights-space around a switching threshold: a - bottom), they result in non-interfering activation patterns (b - bottom), and effectively, a feature-wise separable effect on the marginal loss (c - bottom).

with the number of modalities. It implies that predictive features of some modalities cannot be learned without also including noisy features from the other modalities. The noisy features then effectively suppress the predictive value of the modality that they come from, leading to their observed collapse in the fused representation, a process we formalize in Theorem 1.

As depicted in Figure 1, in the optimization landscape, collapse corresponds to a suboptimal solution such that any step around it along the dimension of entanglement (which is the only available dimension for optimization in the given state), would lead to a simultaneous denoising of one modality and the forgetting of predictive features from the other. If the latent factors underlying the modalities are sufficiently complementary, we show that cross-modal predictive–predictive feature entanglements are less likely to occur than predictive–noisy entanglements (Lemma 3) – so, we mainly focus on the latter in this work.

We find that this cross-modal entanglement of features happens due to faulty neural capacity allocation (Scherlis et al., 2022) among modalities during the optimization of the fusion head (illustrated in Figure 2), which we observe, in Lemma 2, to be a result of the well-known low-rank simplicity bias of neural networks (Huh et al., 2023) limiting the rank of the gradient updates received at any given layer.

Consequently, through Theorem 2, we arrive at the result that this gradient-rank bottleneck forces SGD to parameterize the fusion head neurons in a polysemantic manner.

Interestingly, we observe in Theorem 3, that knowledge distillation into the modalities that collapse, from the ones that survive, implicitly averts cross-modal polysemantic entanglements. It does so by freeing up rank bottlenecks at the level of the student encoders. As a result, as again shown in Figure 1, the noisy features of the modalities that would otherwise collapse, are allocated dedicated dimensions in the latent space, which the fusion operator can then leverage to denoise the output representations. This allows for the complete incorporation of predictive features from all the modalities without any noisy interference, thereby preventing collapse.

Under the condition of identifiability (Gulrajani & Hashimoto, 2022) of modality-specific causal factors up to equivariances of the underlying mechanisms (Ahuja et al., 2022), we propose an algorithm called Explicit Basis Reallocation (EBR), which automatically identifies cross-modal feature entanglements and learns independent denoising directions in the latent space to counteract their hindrance on empirical risk minimization. Consequently, the concrete feature-to-basis mapping across modalities obtained from EBR can be used to identify suitable substitution candidates

for dealing with missing modalities at test-time.

To summarize, we (i) provide a theoretical understanding of modality collapse based on polysemantic neurons in the fusion head leading to unwanted cross-modal entanglements, and the low-rank simplicity bias of neural networks; (ii) show that cross-modal knowledge distillation into the modalities undergoing collapse from the ones that survive has an implicit effect of averting modality collapse through disentanglement and denoising, based on which we propose Explicit Basis Reallocation (EBR) for a more systematic disentanglement and denoising of multimodal embeddings; (iii) extensive empirical validation of our theoretical results on multiple standard multimodal benchmarks, with EBR achieving state-of-the-art (SOTA) results in the application of dealing with missing modalities at test time, one of the most challenging tests of an algorithm's robustness to modality collapse.

## 2. Related Works

**Modality Collapse:** One of the earliest reports of modality collapse was in multimodal generative models (Shi et al., 2019; Sutter et al., 2021; Ma et al., 2020), for which Nazábal et al. (2020) hypothesized the phenomenon to be a result of disparities between gradients, which was also later confirmed by Javaloy et al. (2022). Parallely, Wang et al. (2020) showed modality collapse for multimodal classification problems, where they found unimodal models to often outperform multimodal ones. They conjectured that (i) increased capacity of multimodal models leads to overfitting – supported later in Wu et al. (2024); Zhou et al. (2023); and (ii) different modalities generalize at different rates – which also aligns with the findings by Nazábal et al. (2020); Javaloy et al. (2022). Going beyond the hypotheses and conjectures, we aim to develop a rigorous theoretical understanding of modality collapse from the perspective of polysemanticity (Scherlis et al., 2022; Huben et al., 2024; Lecomte et al., 2024) and low-rank simplicity bias (Huh et al., 2023). These theoretical tools have also so far been restricted primarily to unimodal cases, and to the best of our knowledge, we are the first to explore them for explaining failure modes in multimodal learning. Additionally, since our proposed remedies to modality collapse can be leveraged to deal with missing modalities at test time, we provide an extended literature review on this area in Appendix A.

## 3. Collapse Mechanisms and Remedies

**Task Setup:** We study the properties of representations of multimodal data learned by deep modality fusion algorithms. Specifically, based on recent literature (Wu et al., 2024; Zhang et al., 2022; Ma et al., 2022; 2021), we follow the generic setup where samples from a multimodal distribution

$X = \{X_1, X_2, ..., X_m\}, Y$ with $m$ modalities and labels $Y$, first undergo an independent modality-wise encoding through a set of learnable functions $f_1, f_2, ..., f_m$, followed by a learnable modality fusion operator $\varphi : \mathcal{R}^N \to \mathcal{R}^M$, where $N = \dim(f_1) + \dim(f_2) + ... + \dim(f_m)$, and $M$ is any arbitrary integer. Note that since $N$ is finite, $\varphi$ can be considered as a neural network of bounded width and arbitrary depth, as they are known to be universal approximators (Kidger & Lyons, 2020). The output from $\varphi$ is then fed into a classifier head $g : \mathcal{R}^M \to [0, 1]^C$, where $C$ is the number of classes in $Y$, to produce the output label $\hat{y}$. Specifically, the neural representation of and the final prediction on a multimodal sample $\mathbf{x} = \{\mathbf{x}_1, \mathbf{x}_2, ..., \mathbf{x}_m\} \in X$ is obtained as follows:

$$\hat{\mathbf{y}} = g\left(\varphi\left(f_1(\mathbf{x}_1), f_2(\mathbf{x}_2), ..., f_m(\mathbf{x}_m)\right)\right),$$

where $\{\mathbf{x}_1, \mathbf{x}_2, ..., \mathbf{x}_m\}$ are the modality specific instantiations of the sample $\mathbf{x}$. All proofs are provided in Appendix B.

### 3.1. Polysemanticity and Cross-Modal Entanglements

We begin by showing that as the number of modalities increase, the proportion of cross-modal polysemantic neurons, *i.e.*, those that encode features from more than one modality (as opposed to monosemantic neurons which encode exactly one feature from one modality) also increases (Lemma 1). This makes it difficult for the fusion head to independently control the contribution from a given modality without potential destructive interference from others (Theorem 1). It is important to note that the results presented presuppose the occurrence of polysemanticity, *i.e.*, the number of task-relevant features in $X$ is greater than the number of neurons in any layer, something that is most often known to hold in practice (Scherlis et al., 2022).

**Lemma 1** (Cross-Modal Polysemantic Collision). *As the number of modalities increase, the fraction of polysemantic neurons encoding features from different modalities, for a given depth and width, increases quadratically in the number of modalities as follows:*

$$p(\mathbf{w}_p) \geq m(m-1)\frac{(\dim f_{min})^2}{\left(\sum_{i=1}^{m} \dim f_i\right)^2},$$

*where $p(\mathbf{w}_p)$ is the probability of a neuron being polysemantic via superposition, and $f_{min}$ is the modality-specific encoder with the smallest output dimensionality.*

Since we are interested in the fraction of neurons that are cross-modal polysemantic in the space of all polysemantic neurons, $p(\cdot)$ is defined over the space of polysemantic neurons only. Also, Lemma 1 deals specifically with 2-semantic neurons, *i.e.*, those that simultaneously encode 2

features, which is the most likely form of polysemanticity in the combinatorial space of cross-modal polysemanticities for any value of $m$.

**Definition 1** (Conjugate Features). A conjugate feature $\mathbf{z}$ is one that coexists, in a given modality, with another feature $\mathbf{z}^*$ such that at least one of them has some predictive value, but they can semantically cancel each other out when considered in conjunction, *i.e.*,

$$I(\mathbf{z};\mathbf{y}) + I(\mathbf{z}^*;\mathbf{y}) = 0; \ I(\mathbf{z}\mathbf{z}^*;\mathbf{y}) = 0$$

In other words, $\mathbf{z}$ and $\mathbf{z}^*$ noisily interfere with each other.

**Theorem 1** (Interference). *As the number of cross-modal polysemantic collisions increase, the fraction of predictive conjugate features contributing to the reduction of the task loss decreases, resulting in the following limit:*

$$\lim_{p(\mathbf{w}_p)\to 1} \sum_{\forall \mathbf{z}_y \in X} \frac{\partial}{\partial \mathbf{w}_p} \mathcal{L}\left(\varphi(\mathbf{z}_y), \mathbf{y}\right) = 0,$$

*where $\mathbf{z}_y$ denotes predictive conjugate features in $X$.*

The modality facing the above marginal decrease in contribution to the loss reduction across its feature space, is the one that gets eliminated as part of the collapse. Next, we show how this polysemantic interference is a consequence of the low rank simplicity bias in neural networks.

### 3.2. Rank Bottleneck

We establish that with increasing number of iterations, gradient updates in SGD tend to get restricted to a low-rank manifold, the rank of which is proportional to the rank of the average gradient outer product or AGOP (Lemma 2). Consequently, in Theorem 2, we are able to derive an upper-bound of convergence for every weight subspace in a given layer, which gets tighter as the neurons in that layer get increasingly polysemantic (Definition 2). It thus follows that cross-modal polysemantic interference is a result of the low-rank simplicity bias.

**Lemma 2** (Gradient Rank). *The rank of gradient updates across iterations of SGD at layer $l$ is a convergent sequence with the following limit:*

$$\lim_{n\to\infty} \mathrm{rank}(\nabla_l \mathcal{L}_n) \propto \mathrm{rank}\left(\sum_{\mathbf{x}\in X} \nabla\varphi_l(\mathbf{x})\nabla\varphi_l(\mathbf{x})^T\right),$$

*where $\varphi_l(\mathbf{x})$ and $\nabla_l \mathcal{L}_n$ are respectively the output and the gradient of the loss $\mathcal{L}$ at layer $l$ at the $n$-th iteration of SGD, and $X$ is the set of all inputs to layer $l$ across the dataset.*

**Theorem 2** (Polysemantic Bottleneck). *Let $W$ be the weight matrix at a given layer of $\varphi$, and $\mathbf{w} \leq W$ be any subspace in $W$. When the reduction in conditional cross-entropy $H(\mathbf{x};\mathbf{y}|\mathbf{z})$ provided (amount of unique label information held) by each feature is the same, i.e.,*

$I(\mathbf{x};\mathbf{y}|\mathbf{z}_1) = I(\mathbf{x};\mathbf{y}|\mathbf{z}_2) = ... = I(\mathbf{x};\mathbf{y}|\mathbf{z}_k)$, *at any iteration $n$ of SGD, the norm of the difference between $\mathbf{w}$ and the average gradient outer product (AGOP) of the complete weight matrix $W$ is bounded as follows:*

$$\left\| \mathbf{w} - \sum_{x\in X} \nabla\varphi_W(x)\nabla\varphi_W(x)^T \right\| \leq \gamma(\mathbf{w})^{-1/n},$$

*where $\gamma(\mathbf{w})$ is the degree of polysemanticity of $\mathbf{w}$.*

Theorem 2 implies that since the AGOP is known to be the low-rank subspace that $W$ converges to under SGD (Radhakrishnan et al., 2024), the small distance (tighter bound) between the AGOP and subspaces $\mathbf{w}$ with higher $\gamma(\mathbf{w})$ implies that $W$ in fact converges to those subspaces $\mathbf{w}$ with higher degrees of polysemanticity $\gamma(\mathbf{w})$. In other words, SGD is more likely to parameterize $W$ with the low-rank polysemantic neurons than with high-rank monosemantic ones. The implication of this is that, if we specifically consider the noisy-predictive type of cross-modal polysemantic neurons, they will be the first to get eliminated among all cross-modal polysemantic neurons due to the low-rank simplicity bias, as they either do not contribute to loss reduction, or do so negatively. Below we explore ways of breaking this implicit rank bottleneck to circumvent cross-modal polysemantic interference among noisy and predictive features.

### 3.3. Knowledge Distillation Frees Up Rank Bottlenecks

We propose a simple remedy to the cross-modal polysemantic interference that a multimodal fusion model might suffer from under the default training paradigm with SGD. Based on our result in Theorem 3, the solution is to replace the modality-specific encoder of the modality that gets eliminated under collapse, with one that is pretrained via cross-modal knowledge distillation. Specifically, knowledge distillation has to be performed from the modality that survives fusion, to the one that gets ignored under fusion. When more than one modality survives, we experimentally find that distilling in a sequence starting from the weakest and finishing with the strongest provides the best results (Appendix C.3).

**Theorem 3** (Dynamic Convergence Bound). *When the inputs to $\varphi$ are dynamic (for instance, when the unimodal representations are aligned via cross-modal knowledge distillation) under some distance metric $d$, then at any iteration $n$ of SGD, the norm of the difference between $\mathbf{w}$ and the AGOP of $W$ is bounded as follows for all modalities $i, j \in M$ and datapoints $\mathbf{x} \in X$:*

$$\lim_{d(\tilde{\mathbf{x}}_i, \tilde{\mathbf{x}}_j)\to\epsilon} \left\| \mathbf{w} - \sum_{\mathbf{x}\in X} \nabla\varphi_W(\mathbf{x})\nabla\varphi_W(\mathbf{x})^T \right\| \leq \kappa^{-1/n},$$

*where $\tilde{\mathbf{x}}_i, \tilde{\mathbf{x}}_j = f_i(\mathbf{x}_i), f_j(\mathbf{x}_j)$, $\kappa$ is a constant for a given depth proportional to the AGOP along the entire weight matrix $W$ at that depth, $\epsilon$ is the maximum permissible bound*

*on the distance between any pair of modality-specific encodings, and both $W$ and $\mathbf{w}$ are functions of $\mathbf{x}_i$ and $\mathbf{x}_j$, as they result from backpropagation on their predictions on $X$.*

As per Theorem 3, as the representations from different modalities get closer to each other under the distance metric $d$, which is what effectively happens during cross-modal knowledge distillation, the proportion of monosemantic neurons in $W$ increases. This results in the AGOP of $W$ diverging away from its polysemantic subspaces $\mathbf{w}$. In other words, knowledge distillation implicitly disentangles the cross-modal interferences by freeing the rank bottleneck and encouraging necessary monosemanticity, allowing for independent, modality-wise denoising of features along novel dimensions. The intuition behind this observation is graphically illustrated in Figure 3.

### 3.4. Explicit Basis Reallocation

Although knowledge distillation facilitates independent denoising of modality-specific representations in the fusion head by freeing up rank bottlenecks, the processes of disentanglement and denoising are implicit and hence, slow. We leverage our learnings about the disentanglement and denoising dynamics of knowledge distillation and use them as a set of inductive biases to design an algorithm for Explicit Basis Reallocation, which addresses the problem in a significantly more controlled and efficient manner.

All modifications for EBR are restricted at the level of the unimodal encoders, and we do not alter the fusion operator in any way, which makes it agnostic of the choice of the fusion operator. We introduce a simple encoder-decoder head $h_i \cdot h_i^{-1}$ on top of each modality specific encoder such that the unimodal encoding of each modality $i$ can be specified by the function $f_i = \bar{f}_i \cdot h_i \cdot h_i^{-1}$. For notational convenience, let $g_i = \bar{f}_i \cdot h_i$. We also introduce a modality-discriminator network $\psi$ that is trained on $g_i(\mathbf{x})$ to predict the modality labels. $h$, $h^{-1}$ and $\psi$ are simple two-layer MLPs, and hence add minimal parameter overhead. Jointly, we optimize the following two criteria:

$$\mathcal{L}_{\text{md}} = \sum_{i=1}^{m} \mathcal{L}_{\text{CE}}(\psi(g_i(\mathbf{x})), m); \; \mathcal{L}_{\text{sem}} = \mathcal{L}_{\text{CE}}(\hat{\mathbf{y}}, \mathbf{y}),$$

where $\mathcal{L}_{\text{md}}$ and $\mathcal{L}_{\text{sem}}$ respectively stand for the modality discrimination loss and the semantic loss (of the final multimodal prediction) respectively. The modality-specific parameter sets are updated as follows in each iteration of SGD:

$$\psi \leftarrow \psi - \nabla_{\psi}\mathcal{L}_{\text{md}}$$
$$g_i \leftarrow g_i - \nabla_{g_i}\mathcal{L}_{\text{sem}} + \nabla_{g_i}\mathcal{L}_{\text{md}}$$
$$h_i^{-1} \leftarrow h_i^{-1} - \nabla_{h_i^{-1}}\mathcal{L}_{\text{sem}}$$

**Theoretical Rationale:** The maximization of $\mathcal{L}_{\text{md}}$ by $g_i$

brings all the modalities within the $\epsilon$-neighborhood under $d$ specified in Theorem 3, implementing an explicit disentanglement of noisy and predictive features. The adversarial updates to $\psi$ and $g_i$ are continued until the final multimodal prediction loss $\mathcal{L}_{\text{CE}}(\hat{\mathbf{y}}, \mathbf{y})$ decreases, so as to retain the underlying causal factors that are identifiable (Gulrajani & Hashimoto, 2022), alongside modality-specific, semantically relevant features (Chaudhuri et al., 2024) that arise out of equivariances shared by the underlying causal mechanisms (Ahuja et al., 2022). Projecting $g_i(\mathbf{x})$ back into the original dimensionality of $\bar{f}_i$ via $h_i^{-1}$ leads to a denoised representation that utilizes the compete output basis of $\bar{f}_i$ for representing the predictive features of modality $i$, resulting in increased monosemanticity.

## 4. Experiments

**Datasets and Implementation Details:** We choose the MIMIC-IV (Johnson et al., 2023) and avMNIST (Vielzeuf et al., 2018) datasets for our experiments. For MIMIC-IV, we follow the same settings as (Wu et al., 2024) and that of (Wang et al., 2023; Ma et al., 2021) for avMNIST. We use Tian et al. (2020) as our cross-modal knowledge distillation (KD) algorithm of choice applied on top of MUSE (Wu et al., 2024), which also serves as our multimodal baseline for comparing EBR with SOTA. Due to space constraints, we report the results on MIMIC-IV in the main manuscript and defer those on avMNIST to Appendix C.2.

### 4.1. Cross-Modal Polysemantic Interference

**Objective and Settings:** We validate our theory on cross-modal polysemantic interference (Section 3.1) by studying the impact of the unimodal encoder corresponding to the modality that gets eliminated under fusion, on the minimization of the semantic loss. The results are depicted in Figure 4. The multimodal prefix is the modality-specific encoder of the modality that gets eliminated due to collapse. The red curve represents its semantic loss during multimodal training, computed via linear evaluation on its representation. The unimodal baseline is the same encoder, but is additionally optimized to retain unimodal semantic classification performance. Therefore, although both encoders have the same architecture and receive inputs from the same modality, the multimodal prefix only receives gradient updates through the fusion head, whereas the unimodal baseline also directly optimizes the semantic loss.

**Observations and Analyses:** As predicted by Lemma 1, as the number of modalities increase, the number of polysemantic features in the downstream fusion head also increases. Now, since polysemantic features bottleneck the fusion head (Theorem 2) due to rank-constrained gradient updates (Lemma 2), backpropagated gradients through

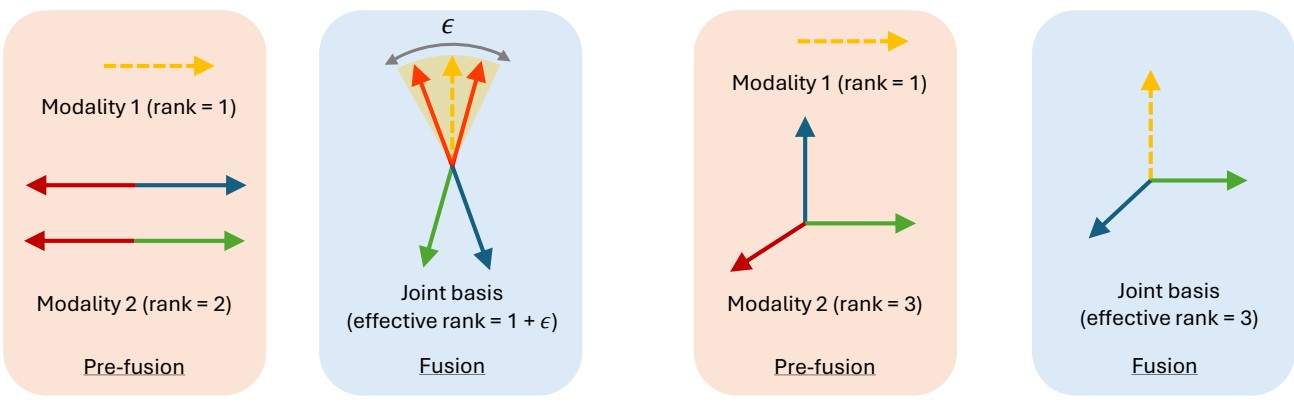

**(a) Cross-Modal Interference due to Rank Bottleneck**   **(b) Rank Bottleneck Free-up via Basis Reallocation**

*Figure 3.* Illustration of modality collapse due to rank bottlenecks enforcing cross-modal polysemantic interference (a), and how freeing up such bottlenecks via basis reallocation can facilitate the elimination of noisy features (red) by encouraging monosemanticity (b).

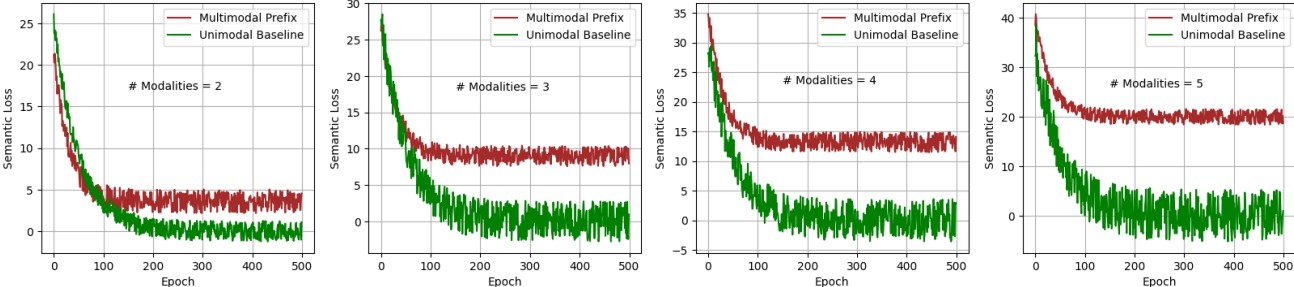

*Figure 4.* MIMIC-IV: Semantic loss curve during training with increasing number of modalities. The multimodal prefix is the semantic loss (linear) evaluation on the modality-specific encoder corresponding to the modality that gets eliminated during multimodal training. The unimodal baseline is the same encoder, but is additionally optimized to minimize its unimodal semantic loss.

the fusion head into the multimodal prefix also get rank-constrained, forcing it to allocate fractional capacities to features that would otherwise have been monosemantically represented. This makes the predictive features harder to decode, leading to the observed gap between the two curves. Since the unimodal model also directly minimizes its own semantic loss, it has a much lower possibility of cross-modal interference, allowing it to successfully perform the necessary capacity allocations, leading to lower loss values. As the number of modalities increase, the gap between the unimodal baseline and the multimodal prefix also increases, aligning with the conclusions of Lemma 1 and Theorem 1.

### 4.2. Presence of Rank Bottlenecks

**Objective and Settings:** We empirically validate our theory linking cross-modal polysemantic interference with the low-rank simplicity bias of neural networks (Section 3.2) by looking at the relationship between the rank of the multimodal representation and the amount of upweighting ($\beta$) needed to force the multimodal model to incorporate the modality that it would otherwise eliminate under collapse. The results are visualized in Figure 5 (a) and (c). The default

setting (w/o KD or w/o EBR) corresponds to the vanilla multimodal model, and the unimodal baseline refers to the rank of the representation learned by the unimodal encoder when trained in a standalone manner to minimize the semantic loss without any multimodal fusion.

**Observations and Analyses:** As the value of $\beta$ (the strength of the modality that gets eliminated by default) is increased, the multimodal rank can be seen to decrease very fast in the default setting. It happens particularly rapidly around a critical point ($\beta = 4$), exhibiting a form of phase transition wherein the rank drops to values lower than the unimodal baseline. As the multimodal model is forced to incorporate more of the said modality, it is forced to select its (mostly noisy) features from the polysemantic subspaces that it has already learned (Lemma 2). So, by the virtue of being represented polysemantically, the rank of this feature subspace ends up being much lower than it otherwise would (as depicted by the unimodal baseline). However, this decay in rank is not observed as we free up rank bottlenecks through basis reallocation, via KD or EBR, implying that rank bottlenecks causing cross-modal polysemantic interference is precisely what is at the root of modality collapse. The rank

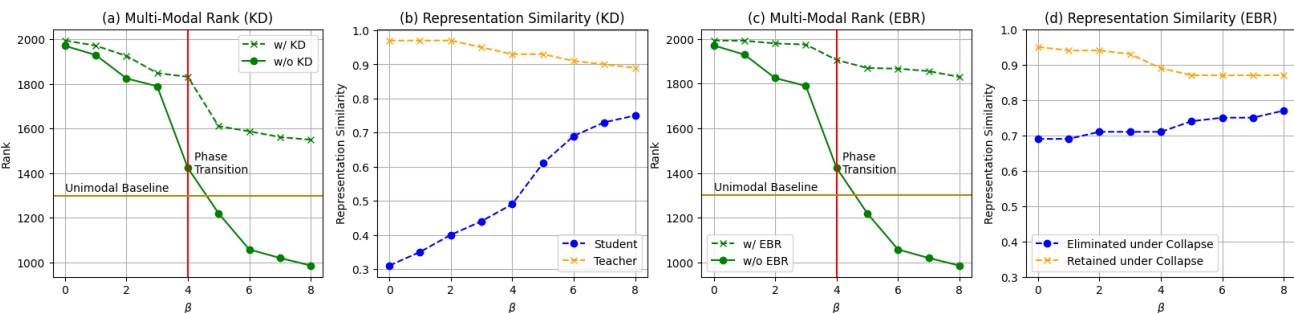

Figure 5. MIMIC-IV: Multimodal rank and representation similarities of modalities with the multimodal representation, under implicit (KD) and explicit (EBR) basis reallocation mechanisms, across different strengths $\beta$ of the modality that gets eliminated under collapse.

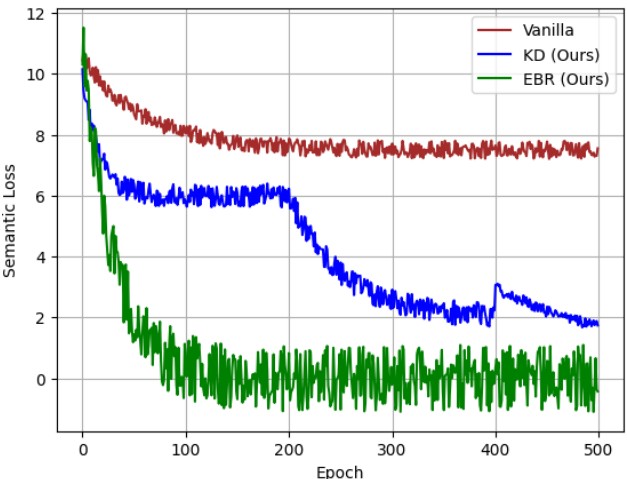

Figure 6. MIMIC-IV: Semantic loss minimization comparison between vanilla multimodal learning and using implicit (KD) and explicit (EBR) basis reallocation.

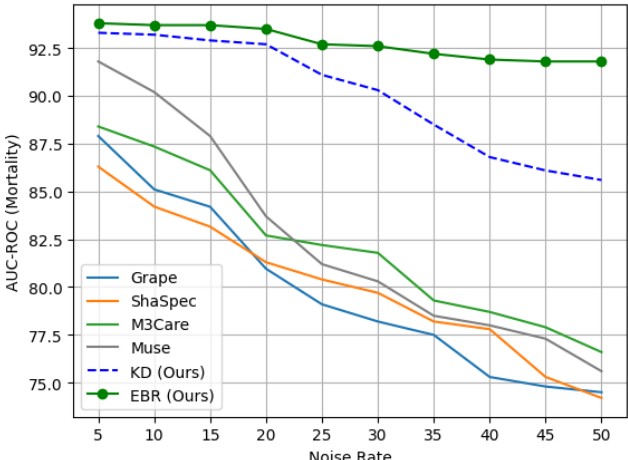

Figure 7. MIMIC-IV: With increasing noise rate, existing approaches suffer from modality collapse due to noisy cross-modal entanglements. With improved strategies of basis reallocation, implicit (KD) or explicit (EBR), robustness to noise and the consequent prevention of modality collapse can be ensured.

of the default multimodal representation being bounded above by that of the unimodal baseline beyond the phase transition around the critical point, is a consequence of the upper-bound presented in Theorem 2.

### 4.3. Effectiveness of Basis Reallocation

Next, we test the effectiveness of basis reallocation (both implicit, via KD, and explicit, via EBR) as a mechanism for freeing up rank bottlenecks to break cross-modal polysemantic interferences, and eventually averting modality collapse. We report our results in Figure 5, Figures 6 and 7, and Table 2, all of which unanimously and unambiguously show the effectiveness of basis reallocation towards preventing modality collapse, confirming the result in Theorem 3.

**Rank and Similarity with the Multimodal Representation:** Figure 5 (a) and (c) provide the most direct evidence that basis reallocation frees up rank bottlenecks, as multimodal representation in both KD and EBR consistently

have a higher rank, while EBR provides a stronger buffer relative to KD against the rank decay occurring around the critical point. Figure 5 (b) and (d) show the representation similarities between the multimodal representation and strongest (teacher) and the weakest (student) modalities. Both the strongest and the weakest modality representations can be seen to more consistently align with the multimodal representation when using EBR, while KD requires some upweighting (by increasing the value of $\beta$) to achieve this alignment. In either case, they indicate that basis reallocation makes the multimodal representation use information from all the modalities, instead of collapsing onto just a subset (strongest) of the modalities.

**Optimization Dynamics:** In Figure 6, we visualize the dynamics of optimizing the semantic loss with or without the implicit (KD) and explicit basis reallocation (EBR) strategies. Generally, we observe that using basis reallocation

| Method | Mortality | | Readmission | |
|---|---|---|---|---|
| | **AUC-ROC** | **AUC-PRC** | **AUC-ROC** | **AUC-PRC** |
| CM-AE (ICML '11) | 0.7873 ± 0.40 | 0.3620 ± 0.22 | 0.6007 ± 0.31 | 0.3355 ± 0.25 |
| SMIL (AAAI '21) | 0.7981 ± 0.11 | 0.3536 ± 0.12 | 0.6155 ± 0.09 | 0.3279 ± 0.15 |
| MT (CVPR '22) | 0.8176 ± 0.10 | 0.3467 ± 0.06 | 0.6278 ± 0.09 | 0.2959 ± 0.05 |
| Grape (NeurIPS '20) | 0.7657 ± 0.16 | 0.3733 ± 0.09 | 0.6335 ± 0.07 | 0.3120 ± 0.11 |
| M3Care (SIGKDD '22) | 0.8265 ± 0.09 | 0.3830 ± 0.07 | 0.6020 ± 0.09 | 0.3870 ± 0.05 |
| ShaSpec (CVPR '23) | 0.8100 ± 0.13 | 0.3630 ± 0.09 | 0.6216 ± 0.10 | 0.3549 ± 0.08 |
| MUSE (ICLR'24) | 0.8236 ± 0.09 | 0.39.87 ± 0.05 | 0.6781 ± 0.05 | 0.4185 ± 0.07 |
| **EBR (Ours)** | **0.8533 ± 0.09** | **0.4277 ± 0.02** | **0.7030 ± 0.05** | **0.4290 ± 0.02** |

*Table 1.* MIMIC-IV: Comparison of average performance with standard deviation across multiple modality missingness rates.

provides improved overall loss minimization as opposed to not using it (vanilla). Specifically, when using implicit (KD), SGD tends to first learn noisy, polysemantic neurons before freeing up rank bottlenecks to denoise them. This leads to multiple step-like structures in the loss trajectory, which could correspond to the saddle geometry of such landscapes. Such geometries are possibly smoothed out into more convex neighborhoods under EBR, providing faster convergence and a more consistent optimization dynamic.

**Denoising Effect of Basis Reallocation:** Theorem 1 posits that cross-modal polysemantic entanglements can be harmful precisely due to the possibility of interference from noisy features. We do a set of experiments where we corrupt the weakest modality during training with additive random uniform noise over a range of noise rates (from 5-50%) and compare SOTA multimodal models with our proposed implicit (KD) and explicit basis reallocation (EBR) mechanisms. We report our findings in Figure 7. Unless explicitly taken care of, existing SOTA models perform notably poorly when the noise rate is increased. Since basis reallocation frees up rank bottlenecks, the novel dimensions can be utilized by SGD for denoising. EBR makes the denoising process explicit through the adversarial training of $\psi$ and $g_i$ to optimize $\mathcal{L}_{\mathrm{md}}$, providing stronger robustness to noise.

| Method | Mortality | | Readmission | |
|---|---|---|---|---|
| | **AUC-ROC** | **AUC-PRC** | **AUC-ROC** | **AUC-PRC** |
| Grape (NeurIPS '20) | 0.8837 | 0.4584 | 0.7085 | 0.4551 |
| + KD | 0.9011 | 0.4620 | 0.7231 | 0.4610 |
| **+ EBR** | **0.9102** | **0.4799** | **0.7488** | **0.4691** |
| M3Care (SIGKDD '22) | 0.8896 | 0.4603 | 0.7067 | 0.4532 |
| + KD | 0.8950 | 0.4700 | 0.7080 | 0.4562 |
| **+ EBR** | **0.8987** | **0.4850** | **0.7296** | **0.4832** |
| MUSE (ICLR'24) | 0.9201 | 0.4883 | 0.7351 | 0.4985 |
| + KD | 0.9350 | 0.4993 | 0.7402 | 0.5066 |
| **+ EBR** | **0.9380** | **0.5001** | **0.7597** | **0.5138** |

*Table 2.* MIMIC-IV: Using knowledge-distilled / EBR backbones for the modality that would otherwise be eliminated by collapse.

**Independence from Fusion Strategies:** Finally, to show

that basis reallocation can be performed agnostic of the fusion strategy, we replace the unimodal encoders of a number of SOTA multimodal models with their knowledge distilled / EBR counterparts and report their performance in Table 2. Irrespective of the fusion strategy, it can be seen that an out-of-the-box improvement in test performance can be attained by these simple replacements, establishing the generic nature of our results.

### 4.4. Fusion with Inference-Time Missing Modalities

As discussed in Section 3.4, after reallocating bases to features via EBR, since the latent factors of $X$ are identifiable up to the equivariances shared by the underlying mechanisms, we leverage this property to substitute missing modalities at test-time with those that are available. Concretely, once training with EBR converges, we proceed as follows: (1) Rank modalities *wrt* their similarities (computed pairwise across all samples) with a reference modality (chosen as the strongest modality in our experiments) in terms of the latent encoding $g_i(\mathbf{x}_i)$; (2) When a modality $i$ of a test sample $\mathbf{x}$ goes missing, choose its substitution candidate as the modality $j$ that is closest to it in the ranked list; (3) Compute the proxy unimodal encoding of $x_i$ as $h^{-1}(g_j(\mathbf{x}_j))$. We validate the importance of ranking based on the EBR latents by benchmarking against other substitution strategies in Appendix C.4.

To evaluate our approach, we adopt the experimental setup of MUSE (Wu et al., 2024), following which we mask out the modalities in the MIMIC-IV dataset with probabilities $\{0.1, 0.2, 0.3, 0.4, 0.7\}$. We then take the average and standard deviation across these missingness rates and report the results in Table 1. It can be seen that close to $3\%$ improvements can be achieved in both Mortality and Readmission prediction AUC-ROC and AUC-PR metrics on top of SOTA, by simply replacing the unimodal encoders of the baseline MUSE with our proposed EBR variants and following the ranking and substitution strategy for dealing with missing modalities detailed above.

## 5. Conclusion and Discussions

We studied the phenomenon of modality collapse from the perspective of polysemanticity and low-rank simplicity bias. We established, both theoretically and empirically, that modality collapse happens due to low rank gradient updates forcing the fusion head neurons to polysemantically encode predictive features of one modality with noisy features from another, leading to the eventual collapse of the latter. This work attempts to reveal that multimodal learning may be plagued in ways that are rather unexpected, and consequently, unexplored, thereby leaving room for a number of improvements and future explorations.

For instance, Theorems 2 and 3 are valid when the reduction in conditional cross-entropy provided (amount of unique label information held) by each feature is the same. It remains to be explored how these results can be extended to the case when such reductions are different across features. We conjecture that (and as also empirically evidenced) EBR turns the otherwise saddle landscape, that is obtained after the rank bottlenecks are freed up by knowledge distillation, into a convex one, enabling smoother and more predictable optimization. Developing an understanding of this could lead to deeper insights into the dynamics of the loss landscape geometry of modality collapse.

## Acknowledgments

We would like to thank the following individuals at Fujitsu Research of Europe for their independent inputs: Mohammed Amer (for discussions on observations of modality collapse in multimodal genomics), Shamik Bose (for inputs on polysemanticity and capacity in neural networks), and Nuria García-Santa (for help with the MIMIC-IV dataset). We would also like to thank the anonymous reviewers for their thorough analysis and detailed feedback that helped clarify and improve various aspects of our work.

## Impact Statement

This paper advances the understanding of modality collapse in multimodal fusion, providing a theoretical foundation and experimental evidence to improve robustness in the presence of missing modalities. By systematically analyzing cross-modal interactions, we demonstrate that mitigating modality collapse enhances performance across diverse applications. In healthcare, our approach can be capable of reliable diagnosis even when hard / expensive to acquire modalities such as imaging or genomics might be missing. In autonomous perception, it could support safer decision-making despite sensor failures. By introducing a scalable and generalizable multimodal learning framework, this work lays the foundation for more robust and deployable AI systems in real-world settings, with the potential to positively impact society. To the best of our knowledge, we are not aware of any negative impacts of this work.

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

## A. Extended Literature Review

**Missing Modalities:** Existing SOTA multimodal fusion approaches do not account for the possibility of missing modalities (Ramachandram & Taylor, 2017; Nagrani et al., 2021; Shi et al., 2021; Chaudhuri et al., 2022; Zhao et al., 2024). Although this limitation was identified in works as early as Ngiam et al. (2011), the pattern of missingness, *i.e.*, which modality(ies) could go missing, were assumed to be known at training time. Later, a number of graph-based techniques (You et al., 2020; Zhang et al., 2022; Wu et al., 2024), including ones that use heterogeneous graphs to model different missingness patterns (Chen & Zhang, 2020), alongside transformer-based (Tsai et al., 2019; Ma et al., 2022), and Bayesian meta-learning (Ma et al., 2021) based approaches attempted to operate without this assumption. Other approaches such as those that facilitate direct interaction among modality-specific raw inputs (Kim et al., 2021; Lee et al., 2023), and ones based on self-supervised domain adaptation (Shen & Gao, 2019), also provided promising results.

Interestingly, it was observed by Ma et al. (2022) that multi-modal representations are strongly dependent on the fusion strategy, and that the optimal way of fusing modalities is dependent on the data, due to which, the authors recommended that fusion strategies should be distribution-specific. However, the limitations introduced by such dependencies was also accounted for in related multimodal learning literature to address the challenge of resource-efficient utilization of modalities, which was tackled through dynamic data-dependent fusion (Xue & Marculescu, 2023). Despite the existence of a number of bespoke techniques for dealing with missing modalities, SOTA approaches such as ShaSpec (Wang et al., 2023), in addition to such algorithms, were also benchmarked against baselines based on GANs (Goodfellow et al., 2020) and autoencoders (Baldi, 2012), which the authors found to be similarly competitive. Although there have been some works that explored the applicability of knowledge distillation to dealing with missing modalities, their purposes have been scoped to addressing issues such as compressing the extra parameter overhead due to multimodal fusion (Dou et al., 2020), or dynamically weighting data points within modalities and contributions from loss terms (Zhou et al., 2023). In this work, we use knowledge distillation as a tool to theoretically study the fundamental processes in optimization that govern modality collapse, and show that it can avoid collapse by implicitly freeing up rank bottlenecks that lead to cross-modal entanglements between noisy and predictive features.

## B. Proofs

**Lemma 1** (Cross-Modal Polysemantic Collision). As the number of modalities increase, the fraction of polysemantic neurons encoding features from different modalities, for a given depth and width, increases quadratically in the number of modalities as follows:

$$p(\mathbf{w}_p) \geq m(m-1)\frac{(\dim f_{\min})^2}{\left(\sum_{i=1}^m \dim f_i\right)^2},$$

where $p(\mathbf{w}_p)$ is the probability of a neuron being polysemantic via superposition, and $f_{\min}$ is the modality-specific encoder with the smallest output dimensionality.

*Proof.* The number of ways any two features can be selected by $\varphi$ from $X$ such that both belong to different modalities is $\geq \binom{m}{2}(\dim f_{\min})^2$, since there are $\binom{m}{2}$ ways of choosing modality-pairs, and there are $\geq (\dim f_{\min})^2$ ways of choosing feature pairs in each such combination. Now, it is these pairs of features that lead to cross-modal polysemantic collisions (through neuron subspaces $\mathbf{w}_p$) during fusion in $\varphi$. Let the ambient dimension of the input to $\varphi$ be $F_{\dim} = \sum_{i=1}^m \dim f_i$. Then, for a given depth and width, the probability that a polysemantic weight subspace would represent features from two different modalities would be:

$$p(\mathbf{w}_p) \geq \binom{m}{2}\frac{(\dim f_{\min})^2}{\binom{F_{\dim}}{2}} = m(m-1)\frac{(\dim f_{\min})^2}{\left(\sum_{i=1}^m \dim f_i\right)^2}$$

This completes the proof of the lemma. □

**Lemma 3** (Entanglement by Feature Type). *If the latent factors underlying the modalities are sufficiently complementary to*

*each other in terms of predicitvity of the label* $\mathbf{y}$, *i.e., for any pair of modalities* $i$ *and* $j$,

$$\sum_p \sum_q \mathbf{z}_i^p \cdot \mathbf{z}_j^q < K,$$

*where* $p$ *and* $q$ *are indices over the latent factors of modalities* $i$ *and* $j$ *respectively, and* $K$ *is a constant, then, the noisy features from one modality are more likely to be entangled with predictive features of another through polysemantic weights in the fusion head, i.e., for any pair of noisy* ($\mathbf{z}_\epsilon$) *and predictive* ($\mathbf{z}_y$) *features from the same modality, the following will hold:*

$$\frac{\sum_{\mathbf{w}} \mathbf{z}_y \cdot \mathbf{w}}{\sum_{\mathbf{w}} \mathbf{z}_\epsilon \cdot \mathbf{w}} \leq 1,$$

*where* $\mathbf{w}$ *denotes weight subspaces representing a different modality in* $\varphi$.

*Proof.* Since noisy features are closer to random, they can get entangled with neurons representing predictive features from any modality if the corresponding neuron allows features up to $(1 - K)$ units of deviations, according to the Johnson–Lindenstrauss lemma (Elhage et al., 2022), *i.e.*, satisfying the following:

$$\mathbf{z}_\epsilon \cdot \mathbf{w} \geq K; \ \mathbf{z}_y \cdot \mathbf{w} < K$$

It implies that the set of noisy features $\mathbf{z}_\epsilon$ would be more closely aligned with $\mathbf{w}$ than the set of predictive features $\mathbf{z}_y$, making the ratio of the sums over all $\mathbf{w} \in \varphi$, of the latter to the former, $\leq 1$.

With the elimination of noisy features, such entanglements following from superposition become less likely among predictive features across modalities, since they would normally require dedicated dimensions with strong orthogonality, *i.e.*, monosemantic neurons. This completes the proof of the lemma. $\square$

**Theorem 1** (Interference). As the number of cross-modal polysemantic collisions increase, the fraction of predictive conjugate features contributing to the reduction of the task loss decreases, resulting in the following limit:

$$\lim_{p(\mathbf{w}_p) \to 1} \sum_{\forall \mathbf{z}_y \in X} \frac{\partial}{\partial \mathbf{w}_p} \mathcal{L}\left(\varphi(\mathbf{z}_y), \mathbf{y}\right) = 0,$$

where $\mathbf{z}_y$ denotes predictive conjugate features in $X$.

*Proof.* Let $\mathbf{z}_\epsilon$ be the noisy conjugate of $\mathbf{z}_y$. As the number of polysemantic collisions increase, so does the proportion of polysemantic neurons, *i.e.*, $\lim_{p(\mathbf{w}_p) \to 1}$. Now, from Lemma 3, we know that as the noisy features are more likely to get into cross-modal polysemantic entanglements, which implies that they would exhibit a higher similarity (dot-product) with the polysemantic subspace $\mathbf{w}_p$. Additionally, since $\mathbf{z}_y$ and $\mathbf{z}_\epsilon$ are conjugate to each other, $\mathbf{z}_y$ would exhibit a low similarity (dot-product) with $\mathbf{w}_p$, activating in the opposite direction as that of $\mathbf{z}_\epsilon$. Again, from Lemma 3, since $\mathbf{z}_\epsilon$ is entangled with $\mathbf{w}_p$, when present in the input, it would always activate. Based on this, the conjugate activation equation (without the non-linearity) of $\mathbf{z}_y$ can be written as:

$$\lim_{p(\mathbf{w}_p) \to 1} \varphi(\mathbf{z}_\mathbf{y}) = \varphi(\mathbf{z}_\mathbf{y}) + \varphi(\mathbf{z}_\epsilon) = \underbrace{\mathbf{w}_p \cdot \mathbf{z}_y}_{\text{large -ve}} + \underbrace{\mathbf{w}_p \cdot \mathbf{z}_\epsilon}_{\text{large +ve}} = 0,$$

meaning that the net activation of $\mathbf{z}_y$ along $\mathbf{w}_p$ is 0, which ultimately implies that:

$$\lim_{p(\mathbf{w}_p) \to 1} \sum_{\forall \mathbf{z}_y \in X} \frac{\partial}{\partial \mathbf{w}_p} \mathcal{L}\left(\varphi(\mathbf{z}_y), \mathbf{y}\right) = 0$$

This completes the proof of the theorem. $\square$

## B.1. Rank Bottleneck

**Definition 2** (Degree of Polysemanticity). We quantitatively define the degree of polysemanticity (Elhage et al., 2022; Scherlis et al., 2022), $\gamma$, of a weight subspace, $\mathbf{w}$, as the ratio of the number of features in the input distribution $X$ encoded in the subspace and the number of dimensions of the subspace, *i.e.*,

$$\gamma(\mathbf{w}) = \frac{|\mathbf{w} \cap X|}{\dim(\mathbf{w})},$$

where $|\mathbf{w} \cap X|$ is the number of features in $X$ that are encoded in $\mathbf{w}$.

Polysemantic neurons lie on a low-rank manifold that the weights of each layer converge to under SGD. So, since all optimization happens along this low-rank polysemantic manifold, it is not possible for $\varphi$ to avert the cancellation effect among conjugate features, of which the noisy counterparts may get encoded as part of a polysemantic neuron.

**Lemma 2** (Gradient Rank). The rank of gradient updates across iterations of SGD at layer $l$ is a convergent sequence with the following limit:

$$\lim_{n \to \infty} \mathrm{rank}(\nabla_l \mathcal{L}_n) \propto \mathrm{rank}\left(\sum_{\mathbf{x} \in X} \nabla \varphi_l(\mathbf{x}) \nabla \varphi_l(\mathbf{x})^T\right),$$

where $\varphi_l(\mathbf{x})$ and $\nabla_l \mathcal{L}_n$ are respectively the output and the gradient of the loss $\mathcal{L}$ at layer $l$ at the $n$-th iteration of SGD, and $X$ is the set of all inputs to layer $l$ across the dataset.

*Proof.* Step 1: The rank of each layer decreases with every iteration of SGD (Galanti et al., 2024). Step 2: Every layer converges to a quantity proportional to the average gradient outer product (Radhakrishnan et al., 2024). □

**Theorem 4** (Depth-Rank Duality (Sreelatha et al., 2024)). *Let $\mathcal{A} = [A_0, A_1, ..., A_n]$ be the attribute subspace of $X$ with increasing ranks, i.e., $\mathrm{rank}(A_0) < \mathrm{rank}(A_1) < ... < \mathrm{rank}(A_n)$, such that every $A \in \mathcal{A}$ is maximally and equally informative of the label $Y$, i.e., $I(A_0, Y) = I(A_1, Y) = ... = I(A_n, Y)$. Then, across the depth of the encoder $\phi$, SGD yields a parameterization that optimizes the following objective:*

$$\underbrace{\min_{\phi, f} \mathcal{L}(f(\phi(X)), Y)}_{ERM} + \min_{\phi} \sum_{l} \left\| \phi[l](\tilde{X}) - \Omega^d \odot \mathcal{A} \right\|_2,$$

*where $\mathcal{L}(\cdot, \cdot)$ is the empirical risk, $f(\cdot)$ is a classifier head, $\phi[l](\cdot)$ is the output of the encoder $\phi$ (optimized end-to-end) at depth $l$, $\|\cdot\|_2$ is the $l^2$-norm, $\odot$ is the element-wise product, $\tilde{X}$ is the $l_2$-normalized version of $X$, $\Omega^d = [\mathbb{1}_{\pi_1(l)}; \mathbb{1}_{\pi_2(l)}; ...; \mathbb{1}_{\pi_n(l)}]$, $\mathbb{1}_\pi$ is a random binary function that outputs 1 with a probability $\pi$, and $\pi_i(l)$ is the propagation probability of $A_i$ at depth $l$ bounded as:*

$$\pi_i(l) = \mathcal{O}\left(\mathrm{rank}(\phi[l]) \, r_i^{-d}\right),$$

*where $\mathrm{rank}(\phi[l])$ is the effective rank of the $\phi[l]$ representation space, and $r_i = \mathrm{rank}(A_i)$.*

**Theorem 2** (Polysemantic Bottleneck). Let $W$ be the weight matrix at a given layer of $\varphi$, and $\mathbf{w} \leq W$ be any subspace in $W$. When the reduction in conditional cross-entropy $H(\mathbf{x}; \mathbf{y}|\mathbf{z})$ provided (amount of unique label information held) by each feature is the same, *i.e.*, $I(\mathbf{x}; \mathbf{y}|\mathbf{z}_1) = I(\mathbf{x}; \mathbf{y}|\mathbf{z}_2) = ... = I(\mathbf{x}; \mathbf{y}|\mathbf{z}_k)$, at any iteration $n$ of SGD, the norm of the difference between $\mathbf{w}$ and the average gradient outer product (AGOP) of the complete weight matrix $W$ is bounded as follows:

$$\left\| \mathbf{w} - \sum_{x \in X} \nabla \varphi_W(x) \nabla \varphi_W(x)^T \right\| \leq \gamma(\mathbf{w})^{-1/n},$$

where $\gamma(\mathbf{w})$ is the degree of polysemanticity of $\mathbf{w}$.

*Proof.* We know that the weights of a neural network when optimized with SGD converge to a value proportional to the Averge Gradient Outer Product (AGOP), which essentially represents those sets of features which when minimally perturbed, produce a large change in the output (Radhakrishnan et al., 2024).

Additionally, SGD with weight decay minimizes the ranks of the weight matrices (Galanti et al., 2024) and that this minimization is more pronounced as we go deeper into the neural network (Huh et al., 2023), as formalized in Theorem 4 by Sreelatha et al. (2024).

In other words, the deeper we go into a network, the more likely it is for the representations to be of a lower rank (Huh et al., 2023; Sreelatha et al., 2024), and that this rank decreases with each successive iteration (Galanti et al., 2024). In other words, for each layer, there is a subspace of a specific rank that is updated through backpropagation, and according to Lemma 2, since the rank of such updates decrease with iterations, the lower the rank of this subspace, the greater its cumulative gradient across iterations, *i.e.*, the more likely it is to be learned by gradient descent and the more likely it is that the weights of the particular layer would converge to this subspace.

If two features equally minimize the empirical risk, and their joint encoding has no local improvement in the minimization of the marginal loss, extending the result by Galanti et al. (2024), SGD on the fusion operator would prioritize the encoding of the modality with the lower rank of the two as follows:

$$\min_{\bar{W}^{\varphi(l)}} \sum_{m \in M} l \left\| \frac{W_m^{\varphi(l)}}{\left\| W_m^{\varphi(l)} \right\|} - W^{\bar{\varphi}(l)} \right\| \le K \cdot (1 - 2\mu\lambda)^{nl}, \tag{1}$$

where $W_m^{\varphi(l)}$ is the subspace of the weight matrix at layer $l$ of the fusion operator $\varphi$ corresponding to the modality $m$, $\mu$ is the learning rate, $n$ is the SGD iteration, and $\bar{W}^{\varphi(l)}$ is the target weight matrix that the fusion operator converges towards at layer $l$ such that:

$$\text{rank}(\bar{W}^{\varphi(d)}) = \sum_{m \in M_c} \text{rank}(\bar{W}^{f_m(o)}),$$

where $\bar{W}_m^{f(o)}$ is the weight matrix at the output layer of the modality-specific encoder $f_m(o)$ of modality $m$ and $M_c$ is the set of modalities that survive collapse. Now, for polysemantic subspaces $\mathbf{w}$, since the degree of polysemanticity $\gamma(\mathbf{w}) > 1$, we can extend Equation (1) as:

$$\min_{\bar{W}^{\varphi(l)}} \sum_{m \in M} d \left\| \frac{W_m^{\varphi(l)}}{\left\| W_m^{\varphi(l)} \right\|} - W^{\bar{\varphi}(l)} \right\| \le K \cdot (1 - 2\mu\lambda)^{nl} \le \gamma(\mathbf{w})^{-1/n} \tag{2}$$

According to the condition $I(\mathbf{x}; \mathbf{y}|\mathbf{z}_1) = I(\mathbf{x}; \mathbf{y}|\mathbf{z}_2) = ... = I(\mathbf{x}; \mathbf{y}|\mathbf{z}_k)$, since basins corresponding to multimodal combinations all lie at the same depth, their empirical risks are essentially the same, and so are the gradients from the ERM term. Now, as a result of modality collapse, we know that one of the basins is steeper than the rest, meaning it has a higher local gradient. Since the empirical risk is constant across all the basins / multimodal combinations, the steepness must come from the rank minimization term. Therefore, the combination with a steep entry must lead to a lower rank solution.

As observed by (Javaloy et al., 2022), no local improvement in the minimization of the marginal loss may be due to conflicting gradients in the *local* parameterizations for the two modalities. Note that this does not imply that the two modalities are globally conflicting. It is only the local encodings of the two that somehow conflict with each other. Specifically, following from Equations (1) and (2) and Lemma 2, the norm of the difference between the polysemantic subspace $\mathbf{w}$ and the AGOP of the ambient weight matrix $W$ can be bound as:

$$\left\| \mathbf{w} - \sum_{x \in X} \nabla \varphi_W(x) \nabla \varphi_W(x)^T \right\| \le K \cdot (1 - 2\mu\lambda)^{nl} \tag{3}$$

Therefore, for polysemantic bases, at any given iteration of SGD, the difference with the AGOP can be more tightly bound than for monosemantic bases. Formally, combining Equations (1) to (3), we have:

$$\left\| \mathbf{w} - \sum_{x \in X} \nabla \varphi_W(x) \nabla \varphi_W(x)^T \right\| \le K \cdot (1 - 2\mu\lambda)^{nl} \le \gamma(\mathbf{w})^{-1/n} \tag{4}$$

In other words a basis formed with polysemantic neurons is more similar to the AGOP than one formed with monosemantic neurons, provided the conditional cross-entropy $H(\mathbf{x}; \mathbf{y}|\mathbf{z})$ reduction provided (amount of unique label information held) by each feature is the same, *i.e.*, $I(\mathbf{x}; \mathbf{y}|\mathbf{z}_1) = I(\mathbf{x}; \mathbf{y}|\mathbf{z}_2) = ... = I(\mathbf{x}; \mathbf{y}|\mathbf{z}_k)$.

This completes the proof of the theorem. □

## B.2. Knowledge Distillation Frees Up Rank Bottlenecks

As described earlier, the cause for collapse is cross-modal interference between noisy and predictive features. Here, we find that knowledge distillation implicitly remedies this problem. Knowledge distillation converges when the noisy and the predictive subspaces have been sufficiently disentangled to the point that the available rank can be assigned completely towards modeling the teacher modality, after effectively having discarded as many of the noisy features as possible. There is some empirical evidence on this from the self-distillation literature (Xie et al., 2019). With the disentangled and denoised representations obtained from knowledge distillation, the causal factors of the previously eliminated modalities can expand (inverse of collapse) the multimodal representation space, utilizing previously unused dimensions for encoding features that effectively reduce the loss. Previously, the effect of the semantically relevant features from the eliminated modalities would not be observable since the superposition with the noisy features would cancel out (when marginalized across all features) any conditional reduction in loss that the causal factors would have induced.

**Theorem 3** (Dynamic Convergence Bound). When the inputs to $\varphi$ are dynamic (for instance, when the unimodal representations are aligned via cross-modal knowledge distillation) under some distance metric $d$, then at any iteration $n$ of SGD, the norm of the difference between $\mathbf{w}$ and the AGOP of $W$ is bounded as follows for all modalities $i, j \in M$ and datapoints $\mathbf{x} \in X$:

$$\lim_{d(\tilde{\mathbf{x}}_i, \tilde{\mathbf{x}}_j) \to \epsilon} \left\| \mathbf{w} - \sum_{\mathbf{x} \in X} \nabla \varphi_W(\mathbf{x}) \nabla \varphi_W(\mathbf{x})^T \right\| \leq \kappa^{-1/n},$$

where $\tilde{\mathbf{x}}_i, \tilde{\mathbf{x}}_j = f_i(\mathbf{x}_i), f_j(\mathbf{x}_j)$, $\kappa$ is a constant for a given depth proportional to the AGOP along the entire weight matrix $W$ at that depth, $\epsilon$ is the maximum permissible bound on the distance between any pair of modality-specific encodings, and both $W$ and $\mathbf{w}$ are functions of $\mathbf{x}_i$ and $\mathbf{x}_j$, as they result from backpropagation on their predictions on $X$.

*Proof.* A weighted general case of the Depth-Rank Duality result by (Sreelatha et al., 2024) follows as a consequence of cross-modal interferences, which can be written as:

$$\underbrace{\min_{\varphi, f} \mathcal{L}(\varphi(X), Y)}_{\text{ERM}} + \alpha \min_{\varphi} \sum_l \text{rank}(\varphi),$$

where $\alpha$ is a function measuring the compactness of the embedding space, in our case, through the pairwise distance between points, $d(\tilde{\mathbf{x}}_i, \tilde{\mathbf{x}}_j)$, computed via a given iterate of $\varphi$, *i.e.*,

$$\alpha = h_\varphi(d(\tilde{\mathbf{x}}_i, \tilde{\mathbf{x}}_j))$$

When cross-modal polysemantic interference happens despite there being extra available dimensions in the ambient space, it indicates a faulty capacity allocation (Scherlis et al., 2022) of the corresponding neurons by SGD, which follows from the Johnson–Lindenstrauss lemma (Elhage et al., 2022). It happens because SGD, by default, performs the aforementioned implicit weighted rank-regularization aside from ERM, with the highest possible value of $\alpha$ such that it encourages the smallest possible rank for a target ERM solution. However, if it is known a priori that the pairwise distances $d(\tilde{\mathbf{x}}_i, \tilde{\mathbf{x}}_j)$ approach some constant neighborhood $\epsilon$, we get:

$$\alpha_1 > \alpha_2 > ... > \alpha_n = h_{\varphi_n}(\epsilon) = A,$$

where $A$ is a constant. Following from this, under the given limit, the rank regularization term in the optimization objective of Depth-Rank Duality gets relaxed as:

$$\lim_{d(\tilde{\mathbf{x}}_i, \tilde{\mathbf{x}}_j) \to \epsilon} \alpha \min_{\varphi} \sum_l \text{rank}(\varphi) = A \min_{\varphi} \sum_l \text{rank}(\varphi),$$

which ultimately implies that for any $\tilde{\varphi}$ inducing pairwise distances $d(\tilde{\mathbf{x}}_i, \tilde{\mathbf{x}}_j) > \epsilon$ will have $\text{rank}(\tilde{\varphi}) < \text{rank}(\varphi)$. As representations from different modalities get closer to each other under the distance metric $d$, which is what effectively happens during cross-modal knowledge distillation, the increase in rank encourages a consequent increase in the proportion

of monosemantic neurons in $W$. This results in the AGOP of $W$ diverging away from its polysemantic subspaces $\mathbf{w}_p$, with the size of such polysemantic subspaces decreasing as follows:

$$\left\| \tilde{\mathbf{w}}_p - \sum_{\mathbf{x} \in X} \nabla \tilde{\varphi}_W(\mathbf{x}) \nabla \tilde{\varphi}_W(\mathbf{x})^T \right\| < \lim_{d(\tilde{\mathbf{x}}_i, \tilde{\mathbf{x}}_j) \to \epsilon} \left\| \mathbf{w}_p - \sum_{\mathbf{x} \in X} \nabla \varphi_W(\mathbf{x}) \nabla \varphi_W(\mathbf{x})^T \right\|,$$

Given the above constraint on the size of polysemantic bases in $W$ as $d(\tilde{\mathbf{x}}_i, \tilde{\mathbf{x}}_j) \to \epsilon$, the size of any feature subspace $\mathbf{w}$, *i.e.*, the number of features that can be encoded by any $\mathbf{w}$ approaches some constant upper bound corresponding to $\epsilon$:

$$\lim_{d(\tilde{\mathbf{x}}_i, \tilde{\mathbf{x}}_j) \to \epsilon} |\mathbf{w} \cap X| = |\mathbf{w}_\epsilon \cap X| = k_{\mathbf{w}},$$

where $|\mathbf{w}_\epsilon|$ is the size of the feature subspace in the neigborhood $\epsilon$. Since both $|\mathbf{w}_\epsilon|$ and $|X|$ are constants, $k_{\mathbf{w}}$ is also a constant. It thus follows that, under a dynamic input space approaching a bounded neighborhood $\epsilon$, the number of features encoded in any polysemantic subspace also gets bounded by $k_{\mathbf{w}}$ as $n \to \infty$. So, the RHS in Theorem 2 can be rewritten as:

$$\lim_{n \to \infty} \gamma(\mathbf{w})^{-1/n} = \left( \frac{|\mathbf{w} \cap X|}{\dim(\mathbf{w})} \right)^{-1/n} = \left( \frac{k_{\mathbf{w}}}{\dim(\mathbf{w})} \right)^{-1/n} = \kappa^{-1/n},$$

where $\kappa = k_{\mathbf{w}} / \dim(\mathbf{w})$ is a constant as both $k_{\mathbf{w}}$ and $\dim(\mathbf{w})$ are constants. Finally, following from the above, when $d(\tilde{\mathbf{x}}_i, \tilde{\mathbf{x}}_j) \to \epsilon$, Equation (4) from the proof of Theorem 2 can be expressed in terms of $\kappa$ as:

$$\lim_{d(\tilde{\mathbf{x}}_i, \tilde{\mathbf{x}}_j) \to \epsilon} \left\| \mathbf{w} - \sum_{x \in X} \nabla \varphi_W(x) \nabla \varphi_W(x)^T \right\| \leq K \cdot (1 - 2\mu\lambda)^{nl} \leq \gamma(\mathbf{w})^{-1/n} = \kappa^{-1/n}$$

This completes the proof of the theorem. □

### B.3. Additional Remarks

**Unequal conditional cross-entropy across features:** According to the condition $I(\mathbf{x}; \mathbf{y} | \mathbf{z}_1) = I(\mathbf{x}; \mathbf{y} | \mathbf{z}_2) = ... = I(\mathbf{x}; \mathbf{y} | \mathbf{z}_k)$, since basins corresponding to multimodal combinations all lie at the same depth, their empirical risks are essentially the same, and so are the gradients from the ERM term. Now, as a result of modality collapse, we know that one of the basins is steeper than the rest, meaning it has a higher local gradient. Since the empirical risk is constant across all the basins / multimodal combinations, the steepness must come from the rank minimization term in Theorem 4. Therefore, the combination with a steep entry must lead to a lower rank solution. When the equality is not met across all features, the low-rank / steepness condition is trivially satisfied by the existence of a lower-dimensional subspace of $\mathbf{z}_i$s that has a lower conditional mutual information $I(\mathbf{x}; \mathbf{y} | \mathbf{z}_i)$, and deriving the upper-bound on the rank in terms of the AGOP is no-longer necessary. The rank of the subspace comprising features with lower relative mutual information could act as a reasonable estimate of the rank of the final weights that SGD would converge to. By considering the condition with the equality, we analyze the boundary case that even when such a subspace with low conditional mutual information cannot be identified, it is possible to upper-bound the rank of the weight matrix.

**Identifying Latent Factors and Substitutability:** If the latent factors are identifiable from the data up to some symmetry of the latent distribution (Gulrajani & Hashimoto, 2022), then the substitutability result also holds up to the actions of that symmetry group. In other words, substitutability is directly contingent on identifiability, *i.e.*, the existence of symmetries in the latent distribution can affect the substitutability of latent factors among modalities.

Segregating predictive and noisy features from the set of latent factors can be done by learning to discover independent causal mechanisms on the aggregate of all modalities (Parascandolo et al., 2018). The degree to which the latent factor representations of the individual modalities can be compressed, *i.e.*, the value of $\epsilon$ in Theorem 3, depends on the size / rank of the invariant (Arjovsky et al., 2019) subspace.

**Intuition behind the Dynamic Convergence Bound:** Every modality consists of both noisy and predictive features. If fusion collapses to a specific modality (target), it means that the modality contains more predictive information and less noise than the rest. Knowledge distillation to align the representations of the other modalities with the target would thus denoise the other modalities, allocating a larger fraction of the feature space of the modality-specific encodings of

such modalities to predictive features. Since noisy features are closer to random, they can get entangled with predictive features from any modality if the corresponding neuron has a slight deviation from perfect orthogonality, according to the Johnson–Lindenstrauss lemma (Elhage et al., 2022). With the elimination of noisy features, such entanglements following from superposition become less likely among predictive features across modalities, since they would normally require dedicated dimensions with strong orthogonality, *i.e.*, monosemantic neurons. Therefore, as the unimodal representations get closer to each other through the implicit denoising mechanism of knowledge distillation, SGD gets increasingly compelled to parameterize the fusion head in a monosemantic manner.

In other words, knowledge distillation implicitly disentangles the cross-modal interferences by freeing rank bottlenecks and encouraging necessary monosemanticity, allowing for independent, modality-wise denoising of features along novel dimensions.

When cross-modal polysemantic weight matrices are rank-bottlenecked, the only solution to minimize the loss further is to allocate the noisy features new, independent dimensions in the latent space. However, this causes an increase in representation rank. To counteract this, knowledge distillation frees up the rank bottlenecks by down-weighting the rank-regularization term.

**Distillation Denoising Conjecture:** Knowledge distillation reduces the weight on the rank-regularization term, freeing up other dimensions for exploration, potentially containing higher rank solutions with more modalities. The rank de-regularization happens due to knowledge distillation having to denoise the student modality in order to align its representation with that of the teacher (Bishop, 1995), as also indicated by our empirical observations in Section 4.3. Given this, we conjecture that knowledge distillation allows the representation of the noisy components of the teacher modality as a transformed version of the student noise, thereby eliminating the need for encoding noisy features from every modality in the neurons encoding the student modality. This can be formally stated as follows:

$$\lim_{f_s(x_s) \to f_t(x_t)} f_t(x_t \hat{\eta}) = g(x_s \hat{\eta}); \quad \forall x_s \in X_s, x_t \in X_t$$

where $X_s$ and $X_t$ are respectively the teacher and student modalities, $f_s$ and $f_t$ are respectively the teacher and student encoders, $\hat{\eta}$ represents the noisy components of a modality, and $g$ is the transformation function relating the student noise with the teacher noise embedding.

**Loss Landscape Geometry:** Once the optimizer converges to the modality-collapsed solution, the value of the loss and the rank of the solution balance each other out. Now, to explore further, the optimizer has to move along a novel direction in the parameter space which leads to a reduction in loss but a simultaneous increase in rank. The reason behind this saddle-geometry is the presence of noisy features from one modality in entanglement with the predictive features from another, which results in an adversarial minimax game between the two. On either side of the saddle point along the unexplored dimensions are predictive features of the former modality which could potentially minimize the task loss, but is not taken into account due to rank regularization. Since, according to Theorem 3, subspaces in $\varphi$ get decreasingly polysemantic when $d(\tilde{\mathbf{x}}_i, \tilde{\mathbf{x}}_j) \to \epsilon$, it implies that knowledge distillation down-weights this regularization by disentanglement and denoising of the bases of the modalities from which the noisy features originate.

**A Note on Figure 2:** Under disentangled polysemanticity, although the noisy and predictive features may still be encoded via the same neuron, they map to different regions in the activation space of the neuron, leading to a feature-wise separable effect. For instance, the coefficients of the predictive features may have a relatively higher magnitude, implying that in order to activate the noisy component of the neuron, the degree of noise in the input would have to be much higher than its predictive counterpart.

## C. Additional Experimental Settings and Results

### C.1. Experimental Settings

**Dataset Details:** MIMIC-IV contains information about 180,000 patients across their 431,000 admissions to the ICU. Following Wu et al. (2024), we use the clinical notes, lab values, demographics (age, gender, and ethnicity), diagnosis, procedure, and medications as the set of input modalities. The task is to perform Mortality (whether the patient will pass away in the 90 days after discharge) and Readmission (whether a patient will be readmitted within the next 15 days following discharge) prediction for a given patient. avMNIST comprises 1500 samples of images and audio, taken from MNIST (Lecun et al., 1998) and the Free Spoken Digits Dataset (Jackson et al., 2018), where the task is to predict the labels of the

input digits from 0 to 9. We adopt the experimental setup of Wang et al. (2023) for avMNIST.

**Implementation Details:** The two hidden layers of $\psi$ have output dimensionalities 512 and 256 respectively. The hidden layers of $h$ have output dimensionalities 1024 and 512 respectively, whereas that of $h^{-1}$ is 512 and 1024. The model was trained for 1200 epochs, with an initial learning rate of 0.01, decayed at a rate of 0.9 every 100 epochs. We interleave between the optimization of $\mathcal{L}_{\mathrm{md}}$ and $\mathcal{L}_{\mathrm{sem}}$ every 10 epochs.

## C.2. Results on avMNIST

Following on from Section 4, we list the empirical results on avMNIST as follows:

- Presence of rank bottlenecks: Figure 10 (a) and (c)

- Effectiveness of basis reallocation:

    - Rank and Similarity with the Multimodal Representation: Figure 10
    - Optimization Dynamics: Figure 9
    - Denoising Effect of Basis Reallocation: Figure 8
    - Independence from Fusion Strategies: Table 4

- Fusion with Inference-time Missing Modalities: Table 3

The analytical conclusions for avMNIST are the same as what is discussed in Section 4, since the patterns of observations are highly consistent between avMNIST and MIMIC-IV. The only experiment not included for avMNIST is the one corresponding to Figure 4 for MIMIC-IV, since avMNIST has only two modalities, and hence, it is not possible to monitor the effect of increasing the number of modalities on the loss curves.

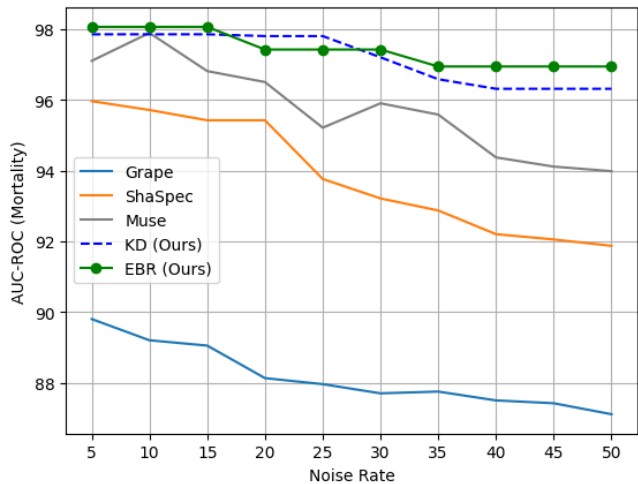

*Figure 8.* avMNIST: With increasing noise rate, existing approaches suffer from modality collapse due to noisy cross-modal entanglements. With improved strategies of basis reallocation, implicit (KD) or explicit (EBR), robustness to noise and the consequent prevention of modality collapse can be ensured.

## C.3. Sequence of Distillation

The results of various strategies for sequencing the teacher modality for cross-modal knowledge distillation are reported in Table 5. Based on these observations, we choose weakest-to-strongest as the sequence to benchmark our KD based implicit basis reallocation mechanism.

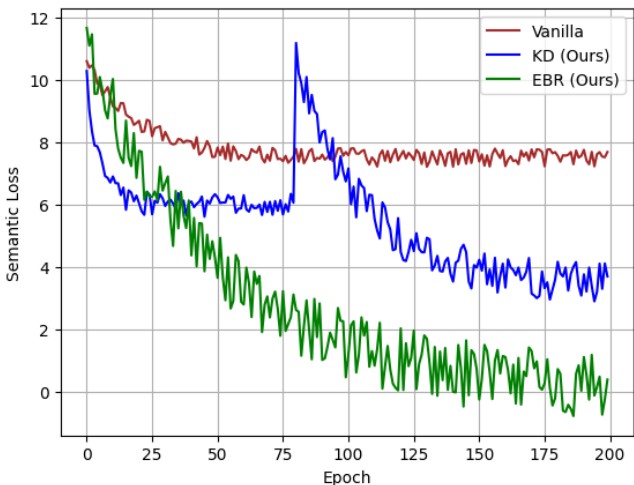

*Figure 9.* avMNIST: Semantic loss minimization comparison between vanilla multimodal learning and using implicit (KD) and explicit (EBR) basis reallocation.

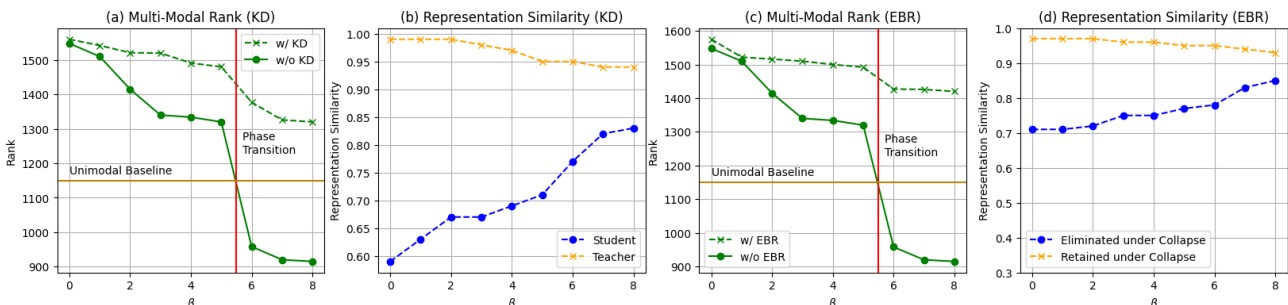

*Figure 10.* avMNIST: Multimodal rank and representation similarities of modalities with the multimodal representation, under implicit (KD) and explicit (EBR) reallocation mechanisms, across different strengths $\beta$ of the modality that gets eliminated under collapse.

| Method | Acc @ Audio Missingness Rate | | | |
|---|---|---|---|---|
| | **95%** | **90%** | **85%** | **80%** |
| Autoencoder (ICMLW'12) | 89.78 | 89.33 | 89.78 | 88.89 |
| GAN (ACM Comm'20) | 89.11 | 89.78 | 91.11 | 93.11 |
| Full2miss (IPMI'19) | 90.00 | 91.11 | 92.23 | 92.67 |
| Grape (NeurIPS'20) | 89.65 | 90.42 | 91.15 | 91.37 |
| SMIL (AAAI'21) | 92.89 | 93.11 | 93.33 | 94.44 |
| ShaSpec (CVPR'23) | 93.33 | 93.56 | 93.78 | 94.67 |
| MUSE (ICLR'24) | 94.21 | 94.36 | 94.82 | 94.93 |
| **EBR (Ours)** | **95.30** | **95.57** | **95.89** | **95.93** |

*Table 3.* avMNIST: Comparison with SOTA on dealing with the missing audio modality across different missingness rates at test time, following the baseline setup of Wang et al. (2023).

## C.4. Baselines for Substitutability

We design the following baselines for comparison against our EBR-based modality substitution approach: zeros, random sampling, nearest-neighbor modality, train set average. Using four (two weak and two strong - diagnosis, lab values, clinical notes, and medication respectively) modalities from MIMIC IV, we report the average with standard deviation of the 15 possible missingness patterns on the AUC-ROC metric for Mortality prediction in Table 6. The target represents the model

| Method | Acc @ Audio Missingness Rate | | | |
|---|---|---|---|---|
| | **95%** | **90%** | **85%** | **80%** |
| SMIL (AAAI'21) | 92.89 | 93.11 | 93.33 | 94.44 |
| + KD | 92.95 | 93.97 | 94.06 | 94.70 |
| **+ EBR** | **93.77** | **94.02** | **94.51** | **94.96** |
| ShaSpec (CVPR'23) | 93.33 | 93.56 | 93.78 | 94.67 |
| + KD | 95.02 | 95.16 | 95.30 | 95.45 |
| **+ EBR** | **95.30** | **95.57** | **95.89** | **95.93** |
| MUSE (ICLR'24) | 94.21 | 94.36 | 94.82 | 94.93 |
| + KD | 94.98 | 95.02 | 95.40 | 95.55 |
| **+ EBR** | **95.02** | **95.26** | **95.61** | **95.70** |

*Table 4.* avMNIST: Using knowledge-distilled / EBR backbones for the modality that would otherwise be eliminated.

| Method | AUC-ROC | AUC-PR |
|---|---|---|
| Strongest only | 0.9196 | 0.4875 |
| Strongest-to-weakest | 0.8651 | 0.4130 |
| Random | 0.9005 | 0.4685 |
| Simultaneous | 0.9102 | 0.4786 |
| **Weakest-to-strongest** | **0.9350** | **0.4993** |

*Table 5.* MIMIC-IV: AUC-ROC and AUC-PR on Mortality Prediction for various sequences of knowledge distillation.

| Method | AUC-ROC | Target |
|---|---|---|
| Zeros | $0.5110 \pm 0.18$ | |
| Random | $0.6139 \pm 0.15$ | |
| Rep-NN | $0.7150 \pm 0.08$ | |
| Late Fusion | $0.6990 \pm 0.06$ | $0.7844 \pm 0.02$ |
| Avg w/o cls | $0.5312 \pm 0.23$ | |
| Avg w/ cls | $0.7396 \pm 0.09$ | |
| **EBR (Ours)** | **$0.7829 \pm 0.05$** | |

*Table 6.* MIMIC-IV: Comparison of the substitutability performance of EBR with baselines. Experiments performed over a subset of 4 (2 strong and 2 weak) input modalities, based on which, the average of 15 possible missingness patterns with standard deviation are reported.

trained on specifically on the subset of modalities that do not go missing, *i.e.*, its performance would always be higher than any baseline since it does not have to deal with the distribution shift that comes from modalities going missing at test time.

### C.5. Multicollinearity

We expect to see increased levels of multicollinearity as the number of modalities increase, if the dimensionality of the representation space remains constant. As correctly conjectured by the reviewer, we would expect multicollinearity to be more pronounced in the deeper layers of the fusion head. The reason behind this is that although there may be dependencies among features across modalities, they may not be exactly linear. As they propagate deeper into the fusion head, it is more likely that those non-linear dependencies would be resolved and linearized in the final representation space prior to classification. Theoretically, the bound in Thm 2 is derived based on the AGOP, i.e., $\mathbf{x} \in X \nabla \varphi_W(\mathbf{x}) \nabla \varphi_W(\mathbf{x})^T$, being a low rank subspace in W (corresponding to an independent set of features), as discussed in (Radhakrishnan et al., 2024), which is also required since one-to-one dimension-to-feature mappings needed to detect the presence of multicollinearity may exist in neural networks (De Veaux & Ungar, 1994). This aligns with the condition for regression multicollinearity that $X^T X$ should be not a full rank matrix.

To empirically confirm this, we calculate the variance inflation factor (VIF) with increasing modalities on our trained representation space. We report the average VIF across features in Table 7. With the increasing number of modalities,

multicollinearity (VIF) increases in all cases. However, basis reallocation encourages cross-modal features to be encoded independently, with the explicit EBR being more efficient in controlling the level of multicollinearity relative to the implicit KD.

| # Modalities | 2 | 3 | 4 | 5 |
|---|---|---|---|---|
| **Vanilla** | 1.15 | 2.68 | 3.51 | 4.70 |
| **w/ KD** | 1.09 | 1.90 | 2.30 | 2.68 |
| **w/ EBR** | 1.05 | 1.26 | 1.32 | 1.55 |

*Table 7.* Average variance inflation factor (VIF) across features in the representation space with increasing number of modalities.

### C.6. Statistical Comparisons

In Table 8 we report the resulting p-values of performing the Wilcoxon rank test with Holm–Bonferroni correction (significance level $\alpha = 0.05$) on the Table 1 results between our proposed EBR and the other baseline methods. The null hypothesis that the proposed EBR and the other models follow the same distribution of AUC-ROC and AUC-PRCs with the chosen missingness rates, were rejected for the both Mortality and Readmission prediction tasks across all baselines, most often, with significantly low p-values, which in all cases, was lower than 0.01. It further provides evidence in support of the uniqueness of EBR in leveraging basis reallocation to free up rank bottlenecks as a novel mechanism to tackle missing modalities.

| Method | Mortality | | Readmission | |
|---|---|---|---|---|
| | AUC-ROC | AUC-PRC | AUC-ROC | AUC-PRC |
| **CM-AE** | 0.0090 | 0.0077 | 0.0065 | 0.0035 |
| **SMIL** | 0.0066 | 0.0053 | 0.0042 | 0.0066 |
| **MT** | 0.0083 | 0.0082 | 0.0077 | 0.0065 |
| **Grape** | 0.0027 | 0.0057 | 0.0058 | 0.0042 |
| **M3-Care** | 0.0079 | 0.0031 | 0.0069 | 0.0039 |
| **ShaSpec** | 0.0085 | 0.0062 | 0.0049 | 0.0075 |
| **MUSE** | 0.0088 | 0.0079 | 0.0086 | 0.0089 |

*Table 8.* P-values of the Wilcoxon rank test with Holm–Bonferroni correction on the Table 1 results between EBR and other baselines.

### C.7. Polysemanticity

Considering the results in Figure 5 (a) and (c), Figure 7, and Section 4.3, since there is no external source of noise in the fusion head, and encouraging monosemanticity through basis reallocation has a denoising effect, the noise that leads to the observed collapse must come from some cross-modal polysemantic interference.

To provide further evidence, we adapt the definition of polysemanticity based on neural capacity allocation from Scherlis et al. (2022) to measure cross-modal polysemanticity as the amount of uncertainty in the assignment of a neuron to a particular modality. We train a two-layer ReLU network on weights from unimodal models to classify which modality the input models are optimized on. Next, we apply this modality-classifier on the weights of our multimodal fusion head and record the average cross-entropy (CE) in its outputs. Higher values of cross-entropy indicate higher levels of cross-modal polysemanticity, since the probability masses are spread out across multiple modalities. In Table 9, we report the results on bi-modal training. The sharply lower relative CE for KD and EBR directly indicate the reduced cross-modal polysemantic interference under basis reallocation.

### C.8. Comparison with Contrastive and Generative Models

**Cross-Modal Polysemantic Interference in multimodal contrastive learning:** We choose GMC (Poklukar et al., 2022) as our candidate multimodal contrastive learning algorithm for analyzing cross-modal polysemantic interference in multimodal contrastive learning. We evaluate GMC by applying their proposed contrastive objective to our baseline representation

| Method | CE |
|--------|-----|
| Vanilla | 5.66 |
| KD | 2.09 |
| **EBR** | **0.59** |

*Table 9.* Empirically measuring cross-modal polysemantic interference as average cross-entropy (CE) in the modality classifier prediction.

learning setting on MIMIC-IV and report the results in terms of the lowest achieved training semantic loss in Table 10. As we can see, trends similar to that of our original setting reported in the main manuscript, in the semantic loss gap between the Multimodal Prefix and the Unimodal Baseline, play out when we perform a contrastive objective based fusion as reported in Poklukar et al. (2022). It further supports the claims in Lemma 1 and Theorem 1 that as the number of modalities increase, the modality undergoing collapse contributes less and less to the downstream representation used to encode the semantics, irrespective of the fusion strategy.

| Number of Modalities | 2 | 3 | 4 | 5 |
|----------------------|-------|-------|-------|--------|
| Multimodal Prefix | 27.68 | 52.90 | 91.20 | 167.30 |
| Unimodal Baseline | 7.97 | 6.55 | 5.33 | 9.55 |

*Table 10.* Lowest achieved training semantic loss with increasing number of modalities in the contrastive setting.

**Rank Bottlenecks in Generative and Contrastive Models:** We choose MMVAE (Shi et al., 2019) as our candidate generative model for analyzing rank bottlenecks. Since the objective of generative modeling is somewhat different from the downstream application that we experimented with, to analyze MMVAE, we performed the experiment on their proposed MNIST-SVHN dataset, while for GMC (Poklukar et al., 2022), since it is for general representation learning, we applied their proposed contrastive objective to our baseline setting on MIMIC-IV. In Table 11, we report the results of our experiment, where the vanilla setting refers to the original model, without KD or EBR.

| Method | $\beta$ | | | | |
|--------|------|------|------|------|------|
| | **0** | **2** | **4** | **6** | **8** |
| **MMVAE (Unimodal Baseline)** | | | 198 | | |
| **MMVAE (Vanilla)** | 477 | 421 | 398 | 110 | 96 |
| **MMVAE + KD** | 482 | 465 | 390 | 298 | 270 |
| **MMVAE + EBR** | 485 | 477 | 431 | 405 | 395 |
| **GMC (Unimodal Baseline)** | | | 1255 | | |
| **GMC (Vanilla)** | 1877 | 1330 | 1146 | 930 | 872 |
| **GMC + KD** | 1905 | 1676 | 1533 | 1427 | 1390 |
| **GMC + EBR** | 1912 | 1825 | 1709 | 1600 | 1588 |

*Table 11.* Representation ranks with increasing $\beta$ in generative (MMVAE) and contrastive (GMC) models.

We can see that in both the generative and contrastive settings, the ranks consistently drop as the strength of the modality undergoing collapse $\beta$ is increased. The drop is sharp around a critical point, where the rank goes below the unimodal baseline, depicting a form of phase transition, a phenomenon also observed in our original experiments (Section 4.2 Observations and Analyses). Finally, the dropping rank can be counteracted by implicit (KD), and even more effectively, by explicit basis reallocation (EBR), which results in a much more stable rank across the range of different values of $\beta$.

