# OpenReview forum: "A Closer Look at Multimodal Representation Collapse"
_ICML.cc/2025/Conference — ICML 2025 spotlightposter_

### Official Review · Reviewer_ZQQA · 2025-02-28

**Overall Recommendation:** 4

**Summary:**

In this paper, the authors contribute with a theoretical understanding of the phenomena of modality collapse in multimodal representation learning model. In particular, the authors show that modality collapse occurs when the predictive features of a given modality become entangled with noise features of another, effectively leading to the collapse of the former. Furthermore, the authors demonstrate that this cross-modal entanglement emerges from faulty neural capacity allocation and that knowledge distillation from the joint encoder into the modality encoder suffering collapse can avert this phenomena. Based on these insights, the authors propose a novel method, Explicit Basis Reallocation (EBR), that promotes the disentanglement and denoising of the multimodal embeddings. The authors evaluate extensively their method across two datasets and highlight how EBR achieves SOTA results on scenarios with missing modalities at test time.

**Claims And Evidence:**

The major claim of the paper is that modality collapse emerges from the entanglement of predictive features from one modality with noisy features of another modality. This claim is significantly supported by both theory and experimental evidence: the authors start by demonstrating in Section 3.1 that the proportion of cross-modal poly semantic neurons increases as the number of modalities increase. This theoretical observation is demonstrated empirically in Section 4.1. Similarly, in Section 4.1 the authors demonstrate that cross-modal interference results from the low-rank simplicity bias. Once again, this claim is empirically evaluated in Section 4.2. Finally. the authors propose Explicit Bias Rallocation (EBR) to address modality collapse. The authors evaluate the effectiveness of EBR extensively in Section 4.3., including results of dealing with missing modalities at inference time.

**Essential References Not Discussed:**

[1] also proposes a method to deal with missing modality information at test time, using a cross-modal contrastive loss.

[1] Poklukar, Petra, et al. "Geometric multimodal contrastive representation learning." International Conference on Machine Learning. PMLR, 2022.

**Experimental Designs Or Analyses:**

The experimental setup presented in Section 4 is sound. Moreover, the authors do a great job in analyzing in depth each result and provide interesting insights.

**Methods And Evaluation Criteria:**

The proposed method appears sound and based on theoretical insights previously discussed in the paper. The evaluation datasets and baselines metrics are also in line with previous literature.

**Other Comments Or Suggestions:**

It would further strengthen the paper if the evaluation presented in Section 4.1 and 4.2 contemplated as well other multimodal representation learning models, such as multimodal variational autoencoders [1-3], where the phenomena of modality collapse has been observed, and other contrastive learning models [4].

[1] - Shi, Yuge, Brooks Paige, and Philip Torr. "Variational mixture-of-experts autoencoders for multi-modal deep generative models." Advances in neural information processing systems 32 (2019).

[2] - Wu, Mike, and Noah Goodman. "Multimodal generative models for scalable weakly-supervised learning." Advances in neural information processing systems 31 (2018).

[3] - Javaloy, Adrián, Maryam Meghdadi, and Isabel Valera. "Mitigating modality collapse in multimodal VAEs via impartial optimization." International Conference on Machine Learning. PMLR, 2022.

[4] - Poklukar, Petra, et al. "Geometric multimodal contrastive representation learning." International Conference on Machine Learning. PMLR, 2022.

**Other Strengths And Weaknesses:**

One significant strength of the paper is the overall quality and completeness of the work: the authors present both interesting theoretical results and an extensive experimental setup.

**Questions For Authors:**

None.


## Post Rebuttal Comment
I thank the authors for their hard work in the rebuttal and for addressing my comments! I maintain my score for now, great job on the paper!

**Relation To Broader Scientific Literature:**

This paper contributes with a theoretical framework to understand modality collapse in multimodal representation learning models. As such, it builds on previous conjectures explored in Javaloy et al., 2022, and Ma et al., 2022. The insights from this work can also be applied to the development of novel multimodal representation learning methods, especially those that deal with large number of modalities.

**Theoretical Claims:**

I reviewed the theoretical claims at a high level but did not verify the correctness of the proofs in detail. The arguments appear well-structured, though a deeper formal verification would be needed to confirm full correctness.

---

> ### Author Rebuttal · Authors · 2025-03-31
>
> We thank the reviewer for recognizing the novelty and thoroughness of our work, as well as pointing us to important adjoining multimodal learning literature that observe modality collapse. Below, we aim to address their concerns, which we will also incorporate in the final version of the manuscript.
>
> **Comparison with generative and contrastive models:**
>
> For the result in Section 4.1, we evaluate [4] by applying their proposed contrastive objective to our baseline representation learning setting on MIMIC-IV and report the results below in terms of the lowest achieved training semantic loss below.
>
> |   | | | Number of Modalities |  |
> | :---- | :---: | :---: | :---: | :---: |
> |  | **2** | **3** | **4** | **5** |
> | **Multimodal Prefix** | 27.68 | 52.90 | 91.20 | 167.30 |
> | **Unimodal Baseline** | 7.97 | 6.55 | 5.33 | 9.55 |
>
> As we can see, trends similar to that of our original setting reported in the main manuscript, in the semantic loss gap between the Multimodal Prefix and the Unimodal Baseline, play out when we perform a contrastive objective based fusion as reported in [4]. It further supports the claims in Lemma 1 and Theorem 1 that as the number of modalities increase, the modality undergoing collapse contributes less and less to the downstream representation used to encode the semantics, irrespective of the fusion strategy.
>
> Unfortunately, we could not perform this experiment on the generative models [1-3], as it would require training each of the generative models from scratch for a number of multimodal combinations, which would take several weeks on our available compute resources, and hence, is infeasible during the rebuttal timeline.
>
> However, we were able to perform the rank evaluation in both the contrastive and generative settings, since pretrained models are available in some cases for the latter. Due to time constraints of the rebuttal period, we chose [1] as our representative generative model. Since the objective of generative modelling is somewhat different from the downstream application that we experimented with, to analyze [1], we performed the experiment on their proposed MNIST-SVHN dataset, while for [4], since it is for general representation learning, we applied their proposed contrastive objective to our baseline setting on MIMIC-IV. Below, we report the results of our experiment, where vanilla setting refers to the original model, without KD or EBR.
>
>
> |  |  |  |  | Beta  |  |  |
> | ----- | :---- | :---: | :---: | :---: | :---: | :---: |
> |  |  | **0** | **2** | **4** | **6** | **8** |
> | | **\[1\] Unimodal Baseline** |  |  | 198  |  |  |
> |  | **\[4\] Vanilla** | 477 | 421 | 398 | 110 | 96 |
> |  | **\[1\] \+ KD** | 482 | 465 | 390 | 298 | 270 |
> |  **Rank** | **\[1\] \+ EBR** | 485 | 477 | 431 | 405 | 395 |
> |  | **\[4\] Unimodal Baseline** |  |  | 1255  |  |  |
> |  | **\[4\] Vanilla** | 1877 | 1330 | 1146 | 930 | 872 |
> |  | **\[4\] \+ KD** | 1905 | 1676 | 1533 | 1427 | 1390 |
> |  | **\[4\] \+ EBR** | 1912 | 1825 | 1709 | 1600 | 1588 |
>
> We can see that in both the generative and contrastive settings, the ranks consistently drop as the strength of the modality undergoing collapse $\beta$ (written as Beta in the table) is increased. The drop is sharp around a critical point, where the rank goes below the unimodal baseline, depicting a form of phase transition, a phenomenon also observed in our original experiments (Sec. 4.2 Observations and Analyses). Finally, the dropping rank can be counteracted by implicit (KD), and even more effectively, by explicit basis reallocation (EBR), which results in a much more stable rank across the range of different values of $\beta$.

---

### Official Review · Reviewer_xQ3V · 2025-03-08

**Overall Recommendation:** 3

**Summary:**

The authors propose a new explanation for the difficult problem of modality collapse. Their argument is that the low-rank bias of neural networks lead them to learn low-rank polysemantic neurons rather than high-rank monosemantic neurons. This is a problem, since as the proportion of cross-modal polysemantic features increases, it prevents the learning of conjugate features that generalize.

**Claims And Evidence:**

* The results in general support the authors claims regarding noisy features and low-rank bias leading to modality collapse.
* The results also highlight the ability of KD/EBR to suppress this behavior and learn higher-rank features.
* However, I think the most crucial result is not very convincing, which is that in Table 1 the downstream performance (AUROC/AUPRC) is not better than the strongest baseline (CM-AE).
* I'm convinced by the evidence that low-rank bias leads to modality collapse. However, there's a large emphasis placed on polysemantic features in the writing, but I don't see this addressed much in the experiments.

**Essential References Not Discussed:**

* It would improve readability to define key terminology such as "polysemantic neurons" for those who are not familiar with the mechanistic interpretability literature.

**Experimental Designs Or Analyses:**

* Figure 4 shows that the ability of the multimodal prefix (the modality that is known to collapse) to predict the target diminishes as the number of modalities increases. It seems like a bit of a logical jump to say that this verifies the claims of Theorems 1 and 2.
* $\beta$ is defined as the "amount of upweighting needed to force the multimodal model to incorporate the modality that it would otherwise eliminate under collapse." I don't see an explanation on what this is exactly. What is being upweighted?
* In Table 1 MUSE is referred to as being the SOTA, but it appears CM-AE is the strongest baseline, and is arguably stronger than EBR as well.

**Methods And Evaluation Criteria:**

* The authors used conventional datasets for evaluating multimodal learning.

**Other Comments Or Suggestions:**

* Some essential details are missing, such as: in the datasets considered, what are the inputs, what are the targets, what is the size of the dataset, etc. It's simply stated as "For MIMIC-IV, we follow the same settings as (Wu et al., 2024)," which is fine, but you should at least include this information in the supplementary material.

**Other Strengths And Weaknesses:**

N/A

**Questions For Authors:**

* In definition 1, are $\mathbf{z}$ and $\mathbf{z}^*$ both in $R^d$? If so, what is $\mathbf{z} \mathbf{z}^*$? Intuitively, I see what you're saying but the clarity could improve if you defined variables and functions more precisely.

**Relation To Broader Scientific Literature:**

* The problem of modality collapse is extensively studied, well-cited in this paper, and the ideas brought by this paper seem to be a novel direction.

**Theoretical Claims:**

N/A

---

> ### Author Rebuttal · Authors · 2025-03-31
>
> We thank the reviewer for noting important gaps in our initial submission. Below, we aim to address them, which will be included in the final version.
>
> **CM-AE:** We apologize for the confusion caused here and we thank the reviewer for pointing out what was an error in our reporting. While we evaluated all the remaining baselines across the 5 missingness rates (Sec. 4.4), we somehow accidentally added the numbers for CM-AE for different random seeds but no missing modality, which is the same as the first row in Tab. 1 of the MUSE paper. Below, we correct this and report the numbers on CM-AE evaluated in the same missing modality setting as that of the others.
>
> |  | Mort. |  | Rdmn. |  |
> | :---- | ----- | ----- | ----- | ----- |
> |  | **ROC** | **PRC** | **ROC** | **PRC** |
> | **CM-AE** | 0.7873 ± 0.40 | 0.3620 ± 0.22 | 0.6007 ± 0.31 | 0.3355 ± 0.25 |
>
> In line with the other baselines, CM-AE, too, performs much worse in the missing modality setting and remains significantly below our proposed EBR.
>
> **Polysemanticity:** Considering the results in Fig. 5 (a) and (c), Fig. 7, and Sec. 4.3. Denoising Effect of Basis Reallocation, since there is no external source of noise in the fusion head, and encouraging monosemanticity through basis reallocation has a denoising effect, *the noise that leads to the observed collapse must come from some cross-modal polysemantic interference.*
>
> To provide further evidence, we adapt the definition of polysemanticity based on neural capacity allocation from Scherlis et al., 2022, to measure cross-modal polysemanticity as the amount of uncertainty in the assignment of a neuron to a particular modality. We train a two-layer ReLU network on weights from unimodal models to classify which modality the input models are optimized on. Next, we apply this modality-classifier on the weights of our multimodal fusion head and record the average cross-entropy (CE) in its outputs. Higher values of cross-entropy indicate higher levels of cross-modal polysemanticity, since the probability masses are spread out across multiple modalities. Below, we report the results on bi-modal training:
>
> |  | CE |
> | ----- | :---: |
> | Vanilla | 5.66 |
> | KD | 2.09 |
> | **EBR** | **0.59** |
>
> The sharply lower relative CE for KD and EBR directly indicate the reduced cross-modal polysemantic interference under basis reallocation.
>
>
>
> **Empirical validation of Thms 1 and 2:**
> We do not claim that Fig. 4, just by itself verifies Thms 1 and 2. Below, we provide a more comprehensive explanation: \
> **Thm 1:** Thm 1 can be factorized as: (i) as the number of modalities increase (ii) the predictive value of the weaker modality decreases. (ii) is validated by Fig. 4, where the gap in semantic loss between the multimodal prefix and the unimodal baseline increases as the number of modalities increase. (i) is validated by Section 4.3, Denoising Effect of Basis Reallocation, which shows that this drop in predictive value is indeed caused due to interference from noisy features. Combined, they imply that as the number of modalities increase, predictive features of some modalities increasingly get entangled with noisy features of another, leading to the collapse of the former. \
> **Thm 2:** Note that by definition, the lower-rank parameterization predicted by Thm 2, is likely to be polysemantic since it has to fit in more features than the number of available dimensions. Fig 5 (a) and (c) establish the decreasing nature of multimodal rank, and Fig 4 establishes the predictivity degradation implying increased noisy polysemanticity due to the former, i.e., decreased rank. This is further discussed in Sec. 4.2 and L351-355, “The rank of the default multimodal representation being bounded above by that of the unimodal baseline beyond the phase transition around the critical point, is a consequence of the upper-bound presented in Thm 2”.
>
>
> **Beta:** We apologize for the ambiguity in the definition of $\beta$. A clearer definition is provided in Section 4.2, stating that $\beta$ “is the strength of the modality that gets eliminated by default”. Specifically, $\beta$ allows for a custom weighting of the modality that would get eliminated by default, and hence, increasing $\beta$, forces the model to incorporate it.
>
> **Terminologies and Dataset Details:** We thank the reviewer for pointing out these missing details, which we will incorporate in the final version.
>
>
> $\mathbf{z}$**,** $\mathbf{z^*}$**, and** $\mathbf{zz^*}$:
> $z$ and $z^*$ are latent factors of arbitrary dimensionality, and $zz^*$ refers to the inner product between $z$ and $z^*$ in the space of latent factors. Since we try to keep our results agnostic of the vector space from which the latent factors originate, we did not concretize $zz^*$ any further. The dimensionality of $zz^*$ would depend on the nature of the inner product of the task-specific latent space. However, we agree that it is worth clarifying this point, which we will do in the final version.

---

### Official Review · Reviewer_BnzK · 2025-03-13

**Overall Recommendation:** 4

**Summary:**

The manuscript introduces an Explicit Basis Reallocation (EBR) approach to mitigate multimodal collapse. It first explains that multimodal collapse is driven by polysemantic neurons—which increase with the number of modalities—leading these neurons to converge into a low-rank polysemantic subspace, ultimately causing collapse. To address this, the manuscript initially proposes using knowledge distillation (KD) from the "strongest" modality to those that are "weakest". Then, it presents EBR as a better alternative that accelerates convergence and eliminates the need for separate modality-specific knowledge distillation. The experimental results on avMNIST and MIMIC-IV indicate that EBR outperforms standard and KD-based training strategies.

## update after rebuttal
The score was increased from 3: Weak Accept to 4: Accept. The authors addressed my concern about Multicollinearity, added statistical comparisons, and improved the connection between theoretical and empirical results.

**Claims And Evidence:**

I think the paper would benefit from linking multimodal collapse to the multicollinearity problem seen in logistic regression. Since the classification head that takes concatenated features from different modalities acts like logistic regression. Explaining how multicollinearity might cause collapse would make the claims more convincing.

**Essential References Not Discussed:**

It would be great to explore related work related to multicollinearity, and inter- and intra-modality dependencies.

**Experimental Designs Or Analyses:**

- Table 1 lacks a statistical comparison between the best and other models. Usually, it requires running Wilcoxon rank test and correction for multiple comparisons (Holm)

**Methods And Evaluation Criteria:**

Yes, the methods and evaluation criteria make sense for the problem. EBR was compared on two multimodal datasets (avMNIST and MIMIC-IV) and versus multiple baselines. Since EBR is a model-agnostic training strategy that is applied on top of existing backbones like MUSE, the approach builds on previous work and the evaluation is well-suited.

**Other Comments Or Suggestions:**

- It will be hard to reproduce experiments since there are no details on the "simple two-layer MLPs", and training schedules. I assume most of it can be found in MUSE, or other backbones, but it is essential to include the details in the appendix.

**Other Strengths And Weaknesses:**

Strengths:
- The idea is interesting from an empirical results perspective
- Multiple baselines
- Two multimodal datasets

Weaknesses:
- Did not feel that the theoretic component was well connected to the empirical results other than rank results.

**Questions For Authors:**

Overall, it is an interesting approach. I will increase the score if you can reduce confusion and clarify the details.

**Relation To Broader Scientific Literature:**

The solutions are similar to the ideas of DCCAE (Wang et al., 2015), where each unimodal encoder has its own additional training unsupervised objective and CCA for capturing joint information. This manuscript uses supervised objectives for each unimodal encoder and fusion joint head. Additionally, it introduces modality-specific encoder-decoder heads, which mimic autoencoder structure. Overall, previous attempts and this strategy promotes the regularization of individual unimodal encoders to ensure that we capture non-shared features from different modalities.

Wang, Weiran, et al. "On deep multi-view representation learning." International conference on machine learning. PMLR, 2015.

Similarly, the idea of inter- and intra-modality dependencies (e.g., Madaan et al., 2024).

Madaan, Divyam, et al. "Jointly Modeling Inter-& Intra-Modality Dependencies for Multi-modal Learning." Advances in Neural Information Processing Systems 37 (2024): 116084-116105.

**Theoretical Claims:**

- Definition 1 is vaguely defined; I is used but not defined as predictive value. Conjugate features are defined but then not used in the discussion of the experiments.

---

> ### Author Rebuttal · Authors · 2025-03-31
>
> We thank the reviewer for taking the time to thoroughly understand our paper and providing important comments, which we believe has helped in significantly solidifying our findings. Below, we provide our response, which we will also incorporate in the final version of our manuscript.
>
> **Multicollinearity:** We indeed expect to see increased levels of multicollinearity as the number of modalities increase, if the dimensionality of the representation space remains constant. As correctly conjectured by the reviewer, we would expect multicollinearity to be more pronounced in the deeper layers of the fusion head. The reason behind this is that although there may be dependencies among features across modalities, they may not be exactly linear. As they propagate deeper into the fusion head, it is more likely that those non-linear dependencies would be resolved and linearized in the final representation space prior to classification. Theoretically, the bound in Thm 2 is derived based on the AGOP, i.e., $\nabla\varphi_W({x}) \nabla\varphi_W({x})^T$, being a low rank subspace in W (corresponding to an independent set of features), as discussed in Radhakrishnan et al., 2024, which is also required since one-to-one dimension-to-feature mappings needed to detect the presence of multicollinearity may exist in neural networks [a]. This aligns with the condition for regression multicollinearity that $X^TX$ should be not a full rank matrix.
>
> To empirically confirm this, we calculate the variance inflation factor (VIF) with increasing modalities on our trained representation space. We report the average VIF across features below.
>
> |  |  | \# Modalities |  |  |
> | :---- | ----- | ----- | ----- | ----- |
> |  | **2** | **3** | **4** | **5** |
> | **Vanilla** | 1.15 | 2.68 | 3.51 | 4.70 |
> | **w/ KD** | 1.09 | 1.90 | 2.30 | 2.68 |
> | **w/ EBR** | 1.05 | 1.26 | 1.32 | 1.55 |
>
> With the increasing number of modalities, multicollinearity (VIF) increases in all cases. However, basis reallocation encourages cross-modal features to be encoded independently, with the explicit EBR being more efficient in controlling the level of multicollinearity relative to the implicit KD.
>
> [a] Veaux and Ungar. “Multicollinearity: A tale of two nonparametric regressions.”, Lecture Notes in Statistics, 1994.
>
> **Definition 1:** We apologize for not clarifying the details. $I(z)$ refers to the mutual information between a feature $z$ and the target label $y$, an abbreviation of $I(z; y)$ for notational minimality. Thm 1 holds in the context of definition 1. The latter is validated through our experiments in Fig. 4,  Sec. 4.3. The observations therein necessitate the existence of conjugate pairs $zz^*$ across modalities, which have the capacity to cancel each other out.
>
> **Statistical Comparisons:** Below we report the resulting p-values of performing the Wilcoxon rank test with Holm–Bonferroni correction (significance level $\alpha$ = 0.05) on the Table 1 results between our proposed EBR and the other baseline methods.
>
> |  Method | Mortality |  | Readmission |  |
> | :---- | ----- | ----- | ----- | ----- |
> |  | **AUC-ROC** | **AUC-PRC** | **AUC-ROC** | **AUC-PRC** |
> | **CM-AE** | 0.0090 | 0.0077 | 0.0065 | 0.0035 |
> | **SMIL** | 0.0066 | 0.0053 | 0.0042 | 0.0066 |
> | **MT** | 0.0083 | 0.0082 | 0.0077 | 0.0065 |
> | **Grape** | 0.0027 | 0.0057 | 0.0058 | 0.0042 |
> | **M3-Care** | 0.0079 | 0.0031 | 0.0069 | 0.0039 |
> | **ShaSpec** | 0.0085 | 0.0062 | 0.0049 | 0.0075 |
> | **MUSE** | 0.0088 | 0.0079 | 0.0086 | 0.0089 |
>
> The null hypothesis that the proposed EBR and the other models follow the same distribution of AUC-ROC and AUC-PRCs with the chosen missingness rates, were rejected for the both Mortality and Readmission prediction tasks across all baselines, most often, with significantly low p-values, which in all cases, was lower than 0.01. It further provides evidence in support of the uniqueness of EBR in leveraging basis reallocation to free up rank bottlenecks as a novel mechanism to tackle missing modalities.
>
> **Connection between theoretical and empirical results:**
> We provide the connections for Thms 1 and 2 in our response to Reviewer xQ3V (Empirical validation of Thms 1 and 2). Here, we provide the same for Thm 3. In Sec. 4.3, we in addition to the results on rank and representation similarity (Fig. 5), we show the difference in the loss landscape geometry between implicit and explicit basis reallocation. Further, we provide evidence that collapse happens specifically due to cross-modal noisy interference, and that basis reallocation, by freeing up rank bottlenecks, allows the new dimensions to be used for denoising, leading to its effectiveness.
>
> **Experimental Details:** \
> $\psi$: 512 -> 256 \
> $h$: 1024 -> 512 \
> $h^{-1}$: 512 -> 1024 \
> \# Epochs: 1200 \
> LR: Initially 0.01 decayed at a rate of 0.9 every 100 epochs \
> We interleave between the optimization of $L_{md}$ and $L_{sem}$ every 10 epochs.

---

> > ### Comment · Reviewer_BnzK · 2025-04-02
> >
> > Thank you for addressing my concerns and questions. I have increased my score to "Accept" to reflect this.

---

### Official Review · Reviewer_NmK3 · 2025-03-21

**Overall Recommendation:** 4

**Summary:**

The paper investigates modality collapse in multimodal learning, where models rely only on a subset of modalities. It shows that this collapse occurs due to entanglement of noisy features from one modality with predictive features from another, leading to suboptimal solutions. The authors propose Explicit Basis Reallocation (EBR) to prevent collapse by reallocating basis vectors in the latent space. Extensive experiments validate the theoretical claims and demonstrate state-of-the-art performance in handling missing modalities.

**Claims And Evidence:**

The claims are well-supported by both theoretical analysis and empirical evidence.

**Essential References Not Discussed:**

N/A

**Experimental Designs Or Analyses:**

Yes. The authors systematically analyze the impact of increasing modalities, noise levels, and missing modalities on modality collapse. The evaluation of both implicit (knowledge distillation) and explicit (EBR) basis reallocation are presented clearly.

**Methods And Evaluation Criteria:**

Yes

**Other Comments Or Suggestions:**

N/A

**Other Strengths And Weaknesses:**

- **Strengths**: The paper provides a comprehensive theoretical analysis and empirical validation of modality collapse, a significant issue in multimodal learning. The proposed EBR algorithm is effective and achieves state-of-the-art results in handling missing modalities.

- **Weaknesses**: The paper assumes that the latent factors are identifiable up to certain symmetries, which might not always hold in practice and could limit the generalizability.

**Questions For Authors:**

- Q1: How do the theoretical results change when the reduction in conditional cross-entropy provided by each feature is not the same across features? Could you provide some insights or preliminary results in this direction?
- Q2: In real-world applications, how would you address the potential issue of non-identifiable latent factors? Are there any practical methods to ensure identifiability up to the required symmetries?

**Relation To Broader Scientific Literature:**

The paper builds on prior work in multimodal learning, specifically addressing the problem of modality collapse. The proposed EBR algorithm contributes to the broader literature by offering new insights into improving the robustness of multimodal models.

**Theoretical Claims:**

I did not check the correctness of the proofs in detail, but the theoretical claims including Lemma 1, Theorem 1, and Theorem 2, appear to be sound based on the provided derivations and explanations.

---

> ### Author Rebuttal · Authors · 2025-03-31
>
> We thank the reviewer for recognizing the theoretical and empirical contributions of our work towards understanding modality collapse and providing valuable feedback. Below, we address their concerns, which we will incorporate in our final version.
>
> **Identifiability:** Indeed there are practical ways of ensuring identifiability of latent factors by ruling out common symmetries. For instance, Gulrajani & Hashimoto, 2022 show that if the CP (Canonical Polyadic) decomposition of the third moment tensor of the distribution of the underlying latent factors is unique and its rank is equal to the dimensionality of the vector space in which the factors lie, then general linear symmetries can be ruled out, ensuring identifiability, for which they also provide an efficient algorithm. Ahuja et al., 2022 also shows that looking at inductive biases of the causal mechanisms instead of the true underlying latent factors is sufficient from identifiable representation learning up to any equivariances shared by the mechanisms. So, our assumption on identifiability up to required symmetries is indeed a widespread one in the literature, and there are many practical methods to ensure this as well.
>
> **Unequal conditional cross-entropy across features:** According to the condition $I(x; y|z_1) = I(x; y|z_2) = ... = I(x; y|z_k)$, since basins corresponding to multimodal combinations all lie at the same depth, their empirical risks are essentially the same, and so are the gradients from the ERM term. Now, as a result of modality collapse, we know that one of the basins is steeper than the rest, meaning it has a higher local gradient. Since the empirical risk is constant across all the basins / multimodal combinations, the steepness must come from the rank minimization term in Theorem 4 (Depth-Rank Duality (Sreelatha et al., 2024)). Therefore, the combination with a steep entry must lead to a lower rank solution.
> When the equality is not met across all features, the low-rank / steepness condition is trivially satisfied by the existence of a lower-dimensional subspace of $z_i$s that has a lower conditional mutual information $I(x; y|z_i)$, and deriving the upper-bound on the rank in terms of the AGOP is no-longer necessary. The rank of the subspace comprising features with lower relative mutual information could act as a reasonable estimate of the rank of the final weights that SGD would converge to.
> By considering the condition with the equality, we analyze the boundary case that even when such a subspace with low conditional mutual information cannot be identified, it is possible to upper-bound the rank of the weight matrix.

---

> > ### Comment · Reviewer_NmK3 · 2025-04-06
> >
> > Thank you for the authors' response, which has resolved my concerns. As a result, I will be upgrading my score to Accept.

---

### Decision · Program_Chairs · 2025-05-01

**Decision:**

Accept (spotlight poster)

**Comment:**

This paper studies the phenomenon of modality collapse in multimodal representation learning models. The authors first provide a theoretical analysis, showing that modality collapse occurs when predictive features from one modality become entangled with noise features from another, effectively suppressing the informative signal of the former. To address this, they propose a novel method called Explicit Basis Reallocation (EBR), which promotes disentanglement and denoising of multimodal embeddings. Extensive experiments support the theoretical findings and demonstrate state-of-the-art performance, particularly in scenarios involving missing modalities.

Reviewers found the paper makes a valuable contribution to the study of modality collapse in multimodal representation learning.